# Cobra Venom Cytotoxins as a Tool for Probing Mechanisms of Mitochondrial Energetics and Understanding Mitochondrial Membrane Structure

**DOI:** 10.3390/toxins16070287

**Published:** 2024-06-25

**Authors:** Edward S. Gasanoff, Ruben K. Dagda

**Affiliations:** 1STEM Research Center, Chaoyang Kaiwen Academy, Beijing 100018, China; edward.gasanoff@cy.kaiwenacademy.cn; 2Belozersky Institute of Physico-Chemical Biology, M.V. Lomonosov Moscow State University, Moscow 119991, Russia; 3Department of Pharmacology, University of Nevada Medical School, Reno, NV 89557, USA

**Keywords:** cobra venom cytotoxins, updated working model of the chemiosmotic theory, non-bilayer lipid phase, cytotoxins, neurodegeneration, bioenergetics, ATP synthase

## Abstract

In this paper, we provide an overview of mitochondrial bioenergetics and specific conditions that lead to the formation of non-bilayer structures in mitochondria. Secondly, we provide a brief overview on the structure/function of cytotoxins and how snake venom cytotoxins have contributed to increasing our understanding of ATP synthesis via oxidative phosphorylation in mitochondria, to reconcile some controversial aspects of the chemiosmotic theory. Specifically, we provide an emphasis on the biochemical contribution of delocalized and localized proton movement, involving direct transport of protons though the F_o_ unit of ATP synthase or via the hydrophobic environment at the center of the inner mitochondrial membrane (proton circuit) on oxidative phosphorylation, and how this influences the rate of ATP synthesis. Importantly, we provide new insights on the molecular mechanisms through which cobra venom cytotoxins affect mitochondrial ATP synthesis, mitochondrial structure, and dynamics. Finally, we provide a perspective for the use of cytotoxins as novel pharmacological tools to study membrane bioenergetics and mitochondrial biology, how they can be used in translational research, and their potential therapeutic applications.

## 1. Introduction

Low-molecular-mass (about 7 kD) cytotoxins, also known as cardiotoxins, are the most abundant proteins in cobra venom [1]. Cytotoxins are amphipathic molecules comprising three β-sheets that conform to the core 3D structure of the protein and are connected to three elongated finger-like loops stabilized via four conserved disulfide bonds [1,2]. It is worth noting that cytotoxins are highly amphiphilic due to the apolar tips of their β-sheet fingers and the polar core region from where the β-sheet fingers are extended in the same direction in space [1,2,3,4]. The apolar and polar regions of cytotoxins are divided by long stretches of highly conserved basic lysine and arginine residues [1,2,3,4]. Cytotoxins exert cytotoxic activity through disintegrating plasma membranes and membranes of cell organelles via binding to lipids and proteins located on the membrane surface [1,2]. However, there is currently no consensus regarding the precise molecular mechanisms through which cytotoxins exert their pharmacological actions, due to the broad diversity of cytotoxin studies that have attempted to explain how cytotoxins target the lipid phase or membrane proteins of cell membranes [1]. Despite the diversity of cytotoxin research, substantial numbers of studies have concluded that the primary target of cytotoxins is the lipid phase of biological membranes [1,2,3,4,5].

Cytotoxins CTI and CTII from venom of the Central Asian cobra Naja naja oxiana have been investigated since half a century ago and are the best characterized cytotoxins from snake venom [6,7]. Interestingly, a few decades ago, other researchers revealed that the amino acid sequences of cytotoxins isolated from venoms of various cobras of the same genera share a high homology [1,3]. Cobra venom cytotoxins are also called cardiotoxins due to their ability to induce cardiotoxicity in the envenomated victim, as evidenced by cardiac arrest [1,3] caused by changes in the Na^+^ and Ca^2+^ concentrations across cell membranes affecting the contractility of heart muscles and aorta [8,9,10]. Indeed, cytotoxins induce the opening of Ca^2+^ channels in membranes of heart muscle cells and neurons, which blocks the inward flow of Na^+^ and K^+^, leading to abnormal currents of ions, which cause heart failure and block the depolarization and repolarization phases of cardiac myocytes [1,3,9]. Interestingly, cytotoxins also affect the functioning of the Na^+^/K^+^ channels of ATPase and integrin receptors [11]. Cytotoxins induce adverse effects on cell-mediated reactions such as hemolysis, myotoxicity, inhibition of platelet aggregation, and activation of apoptosis pathways as well as activation of necrotic cell death, presumably due to alterations in the lipid packing in the cell membrane that induce detrimental effects on the proper functioning of membrane proteins [1,3]. Hence, general cytolysis appears to be the result of cytotoxins’ action on the lipid phase of the membrane. However, it is worth noting that the pharmacological mode of action of cytotoxins depends on the lipid composition of the outer membrane monolayer. In addition, it is important to note that comparative studies have shown that cytotoxins across different species of cobras (Naja atra vs. Naja kaouthia) show variable potencies of cytolytic activity. Their ability to penetrate cell membranes is highly dependent on discrete differences in the amino acid sequences of the N-terminal three-fingered motifs, including the number of lysine and arginine residues. For instance, VC-5 from *N. oxiana* has a higher number of basic residues and possesses a higher potency of cytolytic activity compared with VC-1 [1,2,3,12].

Other pharmacological properties of cytotoxins are associated with their anti-cancer and anti-bacterial activities [1,3,13,14] due to their ability to alter the dynamics and organization of the membranes of cells [1,3,4,14,15,16,17,18,19]. It should be noted that the ability of cytotoxins to affect the organization and fluidity of lipid bilayers depends not only on minor differences in the amino acid sequences of cytotoxins [2,5] in their membrane-binding regions but also on the membrane lipid composition of the target membranes [1,2,18]. For example, cytotoxins CTI and CTII exhibit over 90% amino acid sequence homology [1,2] and can phenocopy the membranotropic activity of the C8 subunit of the F_o_ sector in bovine ATP synthase, which is responsible for translocation of protons through the ATP synthase [20,21]. In addition, both cytotoxins induce aggregation of liposomes and dehydration of neutral phosphatidylcholine-containing membrane surfaces [2,22,23]. However, CTI and CTII show variable binding to phosphatidylcholine membranes enriched with different acidic phospholipids [2]. CTII binds to phosphatidylcholine membranes enriched with any acidic phospholipid to trigger the intermembrane exchange of lipids [2,24,25], membrane fusion [2,22,23,24,25], and the formation of non-bilayer lipid structures [2,20,26], whereas CTI triggers the same effects on phosphatidylcholine membranes enriched with cardiolipin but no other acidic phospholipids [2].

To date, the molecular mechanism through which cytotoxins exert their anti-bacterial activity is not well understood. One proposed mechanism of bactericidal activity entails the insertion of cytotoxins through one monolayer of a bilayer membrane, leading to asymmetric enlargement of the membrane monolayers [25] and eventual collapse of membrane integrity in bacteria [1]. Another proposed mechanism of anti-bacterial activity induced by cytotoxins is based on the formation of membrane pores triggered by the oligomerization of cytotoxins on the cell membrane [27]. The oligomerization of cytotoxins in the lipid phase could be driven by aggregation of acidic phospholipids around the cytotoxins [28]. The oligomerization of cytotoxins in cell membranes may lead to the segregation of membrane regions comprising acidic phospholipid–cytotoxin complexes that lead to the formation of membrane pores. This physiological phenomenon is analogous to perforins released from natural killer cells and cytotoxic T cells in response to bacterial infections [1].

Although cytotoxins exert certain levels of toxicity towards many cell types, cytotoxins exhibit higher cytotoxicity towards cancer cells relative to normal cells [14,19]. This selective cytotoxicity of cytotoxins may be partly explained through the high affinity of cytotoxins to acidic phospholipids [14]. The outer monolayer of healthy cells is made of choline phospholipids including neutral phosphatidylcholine, while amino-containing phospholipids such as neutral phosphatidylethanolamine and acidic phosphatidylserine are located in the inner monolayer of healthy cells [29]. Although phosphatidylcholine is a neutral phospholipid, its positively charged choline group is prominently exposed on the outer membrane surface, which repels basic proteins [30]. The malignant transformation of healthy cells to cancer cells is accompanied by the translocation of acidic phosphatidylserine from the inner membrane leaflet to the outer membrane leaflet of the lipid bilayer. This early, abrupt change in lipid topology is a pathological hallmark in cancer transformation, and phosphatidylserine is commonly recognized as a biomarker for cancer cells [31]. Most cytotoxins, if not all, are strongly attracted to acidic phosphatidylserine on the outer leaflet of cancer cell membranes [1,32], triggering an array of cytotoxicity towards cancer cells ranging from an increase in membrane permeability to cell lysis [1,14,19]. Cytotoxins, which do not cause immediate lysis of cancer cells, tend to translocate from the extracellular space through the plasma membrane and to the cytosolic compartments where cytotoxins target organelles including the nucleus [1], lysosomes [13,33], and mitochondria [34,35,36], eventually triggering apoptosis and necrosis [33,35]. For clarity, the conceptual model via which cytotoxin CTII translocates through the plasma membrane and targets the outer and inner membranes of mitochondria is depicted in Figure 1. This conceptual model is based on the work by [37] and analysis of experimental and in silico data reported in [2,20,36,37,38]. Specifically, as depicted in Figure 1, CTII triggers the formation of an intermembrane junction formed of two plasma membranes (step 1), which subsequently forms an inverted micelle in a single plasma membrane that contains CTII, due to the fission of intermembrane contacts enabling its translocation to the cytosol (step 2). In the cytosol, CTII targets mitochondria via binding to the outer mitochondrial membrane (OMM) to form another intermembrane junction through attracting a portion of the OMM to form the nascent stages of the micelle (step 3). Following the fission of the intermembrane junction, CTII is translocated inside the inverted micelle in OMM (step 4). When the inverted micelle dissolves, CTII is exposed to the intermembrane space. CTII then translocates to the inner compartment of the mitochondria via binding to cardiolipin located in the inner mitochondrial membrane (IMM) to form a junction between OMM and IMM, with CTII located in the center of junction (step 5). Finally, following fission of the junction, CTII “nestles” inside the inverted micelle in IMM (step 6). The proposed steps in the translocation of CTII through plasma membrane, OMM, and IMM are in concordance with prior membrane lipid research, including ^31^P-NMR and ^1^H-NMR spectroscopy studies performed in isolated mitochondrial membranes, model membranes, and molecular dynamics simulations that have provided a rationale for how CTII and CTI interact with mitochondrial membranes [2,20,26,36,37]. Intriguingly, it has been shown that small concentrations of CTI and CTII enhance the production of ATP in isolated mitochondria, whereas higher concentrations of CTI and CTII inhibit production of ATP via disruption of the structural integrity of mitochondrial membranes. In the following sections of this work, we review the latest concepts of cellular bioenergetics and describe how application of cytotoxins as pharmacological tools to study the structure/function of mitochondrial membranes contribute toward understanding molecular details of ATP production in cells.

## 2. Contemporary View of Cell Bioenergetics

The chemiosmotic theory, which was conceptualized and developed by Dr. Peter Mitchel, posits that mitochondria—semi-autonomous organelles—are responsible for the production of most of the energy in cells [39]. A schematic of the complexes in mitochondrial membranes that drive the synthesis of ATP is presented in Figure 2.

All components of the electron transport chain (ETC) are contained in the IMM, which separates the matrix from the intermembrane space. The basic concept of mitochondrial oxidative phosphorylation is based on a coupling between redox processes across all of the four complexes I-IV, the translocation of protons through the ATP synthase, and the synthesis of ATP. The overall coupling can be divided into two distinct processes: redox coupling and proton coupling (Figure 2) [40]. In redox coupling, the exothermic oxidation–reduction reactions are coupled with the translocation of protons against the concentration gradient from the matrix into the intermembrane space, and this coupling involves the four Complexes I through to IV [41]. In the proton coupling, which involves only ATP synthase—Complex V, the movement of protons down the concentration gradient from the intermembrane space into the matrix is coupled with ATP synthesis and this coupling is the most essential part of the oxidative phosphorylation process [40]. There has been a great deal of effort focused on understanding the structural–functional aspects of respiratory complexes I, II, III, IV, and ATP synthase and on elucidating physical–chemical aspects of coupling between the ETC and the ATP synthase, which are meticulously and comprehensively described in the excellent review by Kocherginsky [42]. One of the important findings that came out from these endeavors was the understanding that the respiratory complexes are tightly aggregated in super-complexes [43,44,45] and that the loss of tight aggregation of respiratory complexes leads to a decrease in coupling of the redox reactions with the proton transport and an increase in production of reactive oxygen species (ROS) [43,46]. The aggregation of ATP synthase with respiratory complexes has not been observed [44]; however, ATP synthases form dimers [47] which are needed for creation of cristae [48] and entrapment of protons in the crista tips [49]. In the highly respiring mitochondria, ATP synthases oligomerize to form supramolecular structures of ATP synthase dimers, increasing the rate of ATP synthesis and helping mitochondria to cope with the higher energy demand of respiring cells [50].

In the ETC, electrons are transferred from CI and CII and through CIII to CIV with the assistance of UQ and Cyt *c*, the mobile components of the ETC. The energy released from the transport of electrons through the ETC is used for pumping protons via CI, CIII, and CIV from the matrix into the intermembrane space, where they are released upon the CTIII-catalyzed oxidation of UQH_2_ [51]. This generates proton motif force (PMF) and leads to the oxidation of NADH and succinate, consumption of O_2_, and release of H_2_O (Figure 2).

The Δμ_H_^+^ across the IMM generated via the electron transport coupled with the pumping of protons into the intermembrane space has two components: the electrical component caused by a charge difference across membrane surfaces and the chemical component caused by the difference in concentration of delocalized protons found in bulk water (and other ions) on both sides of the IMM. The difference in concentration of delocalized protons across the IMM creates the osmotic force needed for the movement of delocalized protons through the F_o_ section of ATP synthase to trigger rotation of the F_1_ section of ATP synthase necessary for ATP synthesis. Thus, the Δμ_H_^+^ connects the respiration to oxidative phosphorylation via the ATP synthase.

According to Dr. Mitchell’s chemiosmotic theory [52], building up, maintenance, and utilization of the delocalized Δμ_H_^+^ needed for ATP synthesis requires an impermeable, inert, rigid, and insulating membrane [21,40], a requirement that makes demands of the ultrastructure and molecular organization of the IMM. Since the IMM is a multiply folded membrane that is highly dynamic due to the continuously changing morphology from the bilayer to non-bilayer packing of phospholipids [2,15,20,21,53], it appears that the IMM does not strictly meet the requirements needed to support the Mitchell’s chemiosmotic theory. Hence, in the next section, we review the dynamics and structure of IMM and whether it suits the concepts of chemiosmotic theory. In the current section, we focus on several controversies regarding the chemiosmotic theory which have been raised throughout the past 50 years and which have lasted until today.

One of the most apparent controversies relating to the chemiosmotic theory is regarding the concentration gradient of delocalization of protons in bulk water found on both sides of the IMM, which, according to the chemiosmotic theory, is needed to create a flux of free protons to maintain the PMF. The existence of free (delocalized) protons in bulk water is not possible as a free proton would be immediately solvated to form hydroxonium ion H_3_O^+^ via creation of a coordinate covalent bond with a water molecule [41]. Solvation of delocalized protons taking place near the membrane would disrupt the membrane integrity due to the release of a huge amount of energy from the solvation of free protons. Also, a hypothetical accumulation of free delocalized protons in the intermembrane space, which is a high extended external volume, would greatly dilute the concentration of protons and the entropic component of the PMF would be lost [54]. In addition, a concentration gradient of protons on both sides of the IMM would create unphysiological pH in the intermembrane space and matrix, which theoretically could be detrimental to the structure and functions of biological molecules [2,4,15,40].

Another debated issue in relation to the chemiosmotic theory is the correlation between the membrane potential and the PMF, which are commonly considered as equivalent [55]. It is hypothesized that the membrane potential, which establishes a positive charge on the intermembrane space side of the IMM and a negative charge on the matrix side of the IMM (Figure 2), forces protons move to the matrix through the F_o_ rotor of the ATP synthase to trigger rotation of the F_1_ section required for ATP synthesis [40]. Thus, the proton coupling is realized through the gathering of protons as chemical ions across the coupling membrane to create a driving force for the F_o_F_1_–ATP synthase to promote synthesis. However, the yield of mitochondrial ATP measured does not precisely correlate with the membrane potential created via the gradient of proton concentrations in bulk water on both sides of the IMM [56,57] which raises questions regarding some of the tenets of chemiosmotic theory [58] and suggests that other factors that contribute to ATP synthesis should be considered.

Shortly after the 1978 Nobel Prize in chemistry was awarded to Dr. Peter Mitchell for his contributions to the chemiosmotic theory, there were alternative explanations, mainly provided by Dr. H. Tedeschi, who cast doubt that oxidative phosphorylation was the sole contributor to the transmembrane potential, while Dr. H. Rottenberg defended Dr. Mitchell’s notion of protonated delocalized coupling [59]. Interestingly, the renowned English chemist R. J. Williams proposed three years prior to Dr. Mitchell receiving the Nobel Prize award that protons do not exist in aqueous solution but accumulate on the membrane surface to drive ATP synthesis [60]. This concept was indeed confirmed four decades later on thylakoid membranes from plants, using sophisticated physico–chemical equipment to show that the thylakoid membrane serves as a ‘proton capacitor’ [61], and in HeLa cells through employing a fluorescent pH indicator specific for respiratory Complex III and the F_o_ rotor of ATP synthase. These studies led to the conclusion that it is the movement of protons on the membrane surface, but not in bulk water, that predominantly couples the protons to drive ATP synthesis [62].

Only one year after Dr. Williams’ theory regarding localized proton coupling, which opposed an osmotic transmembrane gradient of protons [60], Dr. Yaguzhinsky’s research group performed a simple yet elegant experiment that entailed inserting purified ATP synthase into an octane–water interface that accumulated protons, which was sufficient to promote ATP synthesis [63]. Thus, in this experiment, the octane–water interface served not only as a proton capacitor but also as a Brønsted acid [40]. About two decades later, it was shown that protons efficiently move on the membrane surface to ATP synthase without incurring major losses in H^+^ in the bulk water [64]. At about the same time, Dr. Yaguzhinsky’s research group demonstrated experimental evidence for the existence of a kinetic barrier to the transfer of protons from the bilayer membrane surface to bulk water, suggesting that a considerable amount of energy is needed for solvation of protons absorbed through the membrane surface [65]. Eighteen years later, Dr. Yaguzhinsky’s research group reported the formation of meta-stable bonds between protons and the mitoplast surface [66]. Interestingly, one year after the same research group showed once again that the mitochondrial membrane acts as a conjugate base when it absorbs protons through its surface and as Brønsted acid when it releases protons as a substrate for ATP synthase [67]. In a recent review by Nesterov and Yaguzhinski [68], the role of protons in enabling conformational changes in the F_1_ subunit and in inducing the phosphorylation reaction (ADP → ATP) was emphasized. That comprehensive review postulated that protons passing through the F_o_ subunit do not leave to the bulk water but form a pH gradient between F_o_ and F_1_ that facilitates proton transfer to F_1_. Mechanistically, the protons passing through the F_o_ subunit bind to carboxyl groups of amino acid residues located within F_1_ to cancel electrostatic attraction between the charged amino acid residues of the rotor and stator, leading to conformational changes in the enzyme [68]. Protonation happens at the start of ATP synthesis and during phosphorylation, while deprotonation stops the rotation of F_o_, which inhibits both ATP synthesis and hydrolysis. Overall, the review listed the key roles of protons, including coupling in ATP synthase, transfer of charge and energy, facilitation of ATP rotation via conformational changes, and direct participation in the reaction of ATP synthesis [68].

Another concern related to the Dr. Mitchell’s chemiosmotic theory relates to the membrane’s permeability to protons. To support the concentration gradient of delocalized protons in bulk water on both sides of the IMM, the IMM should be impermeable to protons. However, the experimental data show that phospholipid membranes are permeable to protons, with permeability coefficients ranging from 10^−3^ to 10^−9^ cm s^−1^ (for comparison, permeability coefficients of phospholipid membranes to ionic solutes range from 10^−12^ to 10^−14^ cm s^−1^) [69]. The leak of phospholipid membranes to protons has been modelled by several groups [69,70,71]. Tepper and Voth provided a theoretical explanation for proton permeability via the formation of a ‘transient membrane spanning aqueous structure’ [70]. In our view, this ‘transient membrane spanning aqueous structure’ could be an inverted micelle responsible for the non-bilayer ^31^P-NMR signal that we observed in model membranes and isolated, native IMM [2,4,15,20,21,26]. Non-bilayer structures are highly dynamic in the IMM, as reported in our previous findings and by others, and are highly permeable to protons. Indeed, high permeability to protons has been also observed in liposomal membranes made of phospholipids from the IMM [72]. The above reports provide compelling experimental evidence that challenges the third postulate of the chemiosmotic theory, which states that the IMM should be impermeable to protons and other ionic species.

Since it has been determined that the IMM is permeable to protons, the question now is how protons move transversely and laterally in the IMM. In a recent review article, Morelli et al. [40] shed some valuable insight regarding a plausible mechanism. Morelli et al. provided two mechanisms of protons moving through a membrane: one mechanism via ‘frozen water’ and another via ‘proton wires through the membrane’. Movement of protons via the frozen water mechanism can occur in the voltage gated proton channel Hv1, in which water molecules are immobilized (frozen) along the channel wall and protons in the form of hydroxonium ions are allowed to pass in the Grotthuss-style hopping mechanism [73,74] to balance the concentrations of H_3_O^+^ ions between the aqueous volumes separated by the IMM. The proton wire passage is also realized through the Grotthuss-style hopping of protons, but not over the frozen water molecules; the protons hop over the amino, carboxyl, and hydroxyl groups of amino acid side chains of Hv1, as described by Nagle and Morowitz [75] and Nagle and Tristram-Nagle [76]. As the frozen water mechanism allows the passage of protons in the form of H_3_O^+^ ions, which implies the generation of a concentration gradient of delocalized H_3_O^+^ ions in bulk water across the IMM, we are inclined to believe that protons move through CI, CIII, and CIV via a proton wire. In addition, protons do not disperse in the bulk water as there is a barrier of water molecules strongly attracted via the dipole–dipole and ion–dipole forces to the membrane surface [40,65]. Thus, protons are directed to move laterally through the polar heads of phospholipids by hopping from one polar head group to another (amino, carboxyl, and carbonyl groups) via consecutive changes in the protonation and deprotonation states [77,78]. Moving laterally through the polar heads of phospholipids, protons reach the F_o_ unit of ATP synthase where they move upward via a proton wire [79,80] to the glutamic residue (Glu 58) of the *c* subunit of the F_o_ rotor [81]. Furthermore, it has been proposed that while F_o_ rotates, protons detach from the Glu 58 and move through the hydrophobic core of the IMM and return to the center of respiratory Complex I (Figure 3) [40]. The protons move through the ends of alkyl chains of lipids in a highly anisotropic environment due to Brownian motion [82].

Additional biochemical research on reconstituted systems revealed a direct transfer of protons from Complex I to ATP synthase [83,84,85] and that the efficiency of proton transfer depended on the proximity of Complex I to ATP synthase [84]. The experimental evidence supports the concept that proton coupling is realized through the movement of protons inside the membrane, while dismissing the idea of a concentration gradient of delocalized protons in bulk water moving across the IMM. Overall, the researchers concluded “that when protons are really transferred through a membrane (i) they are never in the form of free delocalized protons and (ii) they are subject to the Grotthuss-style proton hopping” [86].

A decade and a half ago, aerobic ATP synthesis was discovered in the myelin sheath, which is devoid of mitochondria [87,88,89,90]. In some neurons, the presence of F_1_ subunits of ATP synthase in myelin sheaths was determined by oligomycin titration experiments [88]. Also, the considerable presence of mitochondrial respiratory components in lipid rafts in myelin was discovered [91,92], suggesting the existence of a route for delivering mitochondrial respiratory complexes to the myelin sheath [93,94]. Morelli et al. put forward the idea that mitochondria can provide the necessary apparatus of complex proteins required for synthesizing ATP in extra-mitochondrial regions of the cell [94]. Whereas the majority of mitochondrial proteins are synthesized in the nucleus, the mitochondria have their own DNA, termed mtDNA, which transcribes and synthesizes a limited number of proteins including subunits of respiratory complexes and ATP synthase to drive oxidative phosphorylation (OxPhos). Imaging experiments have demonstrated that mitochondria and ER are closely associated, which suggests that the two function in a coordinated manner to maintain homeostasis and when this association is lost, the bioenergetic capacity is impaired [94]. The close interaction of mitochondria with the ER has been also well documented in other studies [95,96]. These observations assume that mitochondria transfer the OxPhos machinery to the ER, rendering the ER’s ability for aerobic ATP synthesis [91,95,97]. In fact, it has recently been demonstrated that mitochondria generate the necessary vesicles that provide the ER with the necessary OxPhos components [98,99]. However, this phenomenon is not ubiquitous and is restricted to a small number of specialized cell types, as supported by the findings in recent years describing extra-mitochondrial OxPhos in cellular districts including rod outer segment discs [100,101,102,103], platelets [104], cell plasma membrane [105,106,107,108,109], exosomes, and microvesicles [110,111]. Overall, these recent findings strongly suggest the existence of OxPhos transfer machinery to extra-mitochondrial sites via ER [93].

In 1986 and 1992, Gabriel et al. [112] and Surchev et al. [113] employed freeze-fracture X-ray crystallography to show that particles of 8.6 nm, exactly matching the size of the dehydrated F_1_ subunit, moved on both sides of the myelin sheath, suggesting that F_1_ subunits of ATP synthase were exposed on both sides of the myelin sheath. Similar results were observed in rat optic nerves [114]. The bi-facing orientation of F_1_ subunits in the myelin sheath (Figure 4) and in the optic nerve shows a distinct topology and orientation to F_1_ subunits in the IMM, where the F_1_ subunits are exposed on the matrix side only, which, according to the chemiosmotic theory, allows the delocalized proton gradient across the IMM to drive ATP synthesis. Thus, in nerve tissue, the bi-facing orientation of F_1_ subunits does not agree with Mitchell’s concept of delocalized proton coupling but agrees with the concept of localized coupling, in which the proton circuit is built entirely inside the membrane and the proton current is regarded as the charge, in accordance with the Grotthuss mechanism [115]. Noteworthily, additional biochemical studies of ATP synthases reconstituted in liposomes also yielded results supporting the concept of a bi-facing orientation of the F_1_ subunits, in agreement with the localized coupling of protons [116]. Remarkably, localized coupling is also supported by a recently introduced novel concept that links extra-mitochondrial OxPhos in the ER to glucose metabolism coupled to the pentose phosphate pathway in a closed-loop process [99]. In our view, the experimental evidence presented in this review does support localized proton coupling with the protons moving according to the Grotthuss mechanism via the polar heads of phospholipids on the membrane surface. However, we raise the question of the possibility that the movement of protons can occur in a form of pure charge (H^+^) from the *c* subunit of the F_o_ rotor to Complex I via the hydrophobic membrane core. Protons detached from Glu 58 are not likely to enter the low dielectric environment, which is not compatible with the charged entity of a proton. To this end, there must be a special vehicle to transfer protons from the F_o_ rotor to Complex I through the low dielectric environment. In the next section, we review all available studies on the usage of cobra venom cytotoxins as tools for probing the mechanisms of cell bioenergetics and suggest a vehicle that transfers protons in the low dielectric environment inside the membrane from the *c* subunit of the F_o_ rotor to Complex I via the formation of transient non-bilayer structures in the form of inverted micelles.

## 3. Cytotoxins as Tools in Unraveling New Mechanisms of Mitochondrial Bioenergetics

In the previous section of this article, we reviewed the important concepts and controversies relating to the contemporary understanding of cell bioenergetics, except for the role of the non-bilayer lipid phase in modulating some aspects of the structure and function of energy-transducing membranes. In our view, this area of science has not received the necessary attention for over half a century. Nonetheless, reports from the past twenty years have convincingly highlighted the importance of non-bilayer lipid structures in modulating the structure and functions of not only mitochondrial membranes but also thylakoid membranes [2,4,15,20,21,26,36,37,53,117,118,119,120,121,122,123,124]. For the latest views regarding the role of the non-bilayer phase in energy-transducing membranes, we direct readers to peruse the most recent comprehensive review cited below, which describes in detail how non-bilayer structures affect the structural architecture and functions of mitochondrial and thylakoid membranes [21]. In this section of our review, we focus on only the role of non-bilayer structures in mitochondrial membranes, as significant progress on understanding the potential roles of non-bilayer structures in promoting ATP synthesis has been achieved thanks to numerous studies that have probed the interaction of cobra venom cytotoxins with native and model mitochondrial membranes.

The existence of non-bilayer structures in mitochondrial membranes was initially reported for the first time 45 years ago. Using ^31^P-NMR spectroscopy to study the organization of lipid bilayers in intact liver mitochondria, Cullis et al. showed that phospholipids in the inner mitochondrial membrane coexisted as both lamellar and non-bilayer phases at 37 °C [125]. Eleven years later, a review article by Tournois de Kruijff [126] highlighted the importance of the role of non-bilayer structures in mediating the polymorphic transition of phospholipids in mitochondrial and other membranes. Two year later, our research group in collaboration with Dr. Yaguzhinski’s group performed a study that used ^31^P-NMR spectrometry to investigate membrane structure in mitochondrial proteolipids that contained cardiolipin bound to proteins of subunits of the F_o_ unit of ATP synthase isolated from bovine liver. That study revealed the presence of a ^31^P-NMR signal at 6 ppm [127]. This ^31^P-NMR signal was attributed to the existence of non-bilayer packed phospholipids. Interestingly, that was the same ^31^P-NMR signal that was previously observed in multilamellar dispersions of phosphatidylcholine and cardiolipin treated with cardiotoxin CTII [128] and in large unilamellar liposomes made of cardiolipin and phosphatidylcholine and treated with cardiotoxin CTII [28]. Molecular dynamics were performed to study the interaction of cardiotoxin CTII with cardiolipin. These in silico studies suggested that the 6 ppm signal observed in the ^31^P-NMR spectrum may have originated from a non-bilayer comprised of cardiolipin molecules electrostatically attracted to lysine and arginine residues from cardiotoxin CTII [129]. Hence, cytotoxins have become a powerful tool for probing the structural organization of model phospholipid membranes mimicking the phospholipid composition of IMM [2,4,14,128,129,130,131]. Cytotoxins were found to phenocopy the membranotropic activity of the C8 subunit of the F_o_ sector in bovine ATP synthase, responsible for translocation of protons through the F_o_ sector [20,21]. In addition, cytotoxins were found to have a high affinity for binding to cardiolipin with the formation of non-bilayer structures [2,4,20,21,128]. Overall, these studies suggest that cardiolipin has an important role in regulation of the formation of non-bilayer structures, which may influence proton transport at the ATP synthase [20,21].

It has recently been reported that cardiolipin interacts with the inner and outer sides (known also as the inlet and outlet half-channels) of bacterial ATP synthase, which is essential for the hydration of the cavities on each side of the channel, leading to the stabilization of proton transport [132]. In addition, in silico studies have suggested that the binding of cardiolipin near flexible loops of the alpha subunit of ATP synthase facilitates penetration of water molecules into the entrance lacunae and facilitates the movement of protons to *c*Asp61. At the same time, the location of cardiolipin near the entrance of the inlet half-channel contributes to the perimembrane localization of protons, which minimizes pH changes in the periplasm [132]. It was concluded that cardiolipin enhanced the rate of proton movement in the ‘frozen’ water molecules in the half-channels via the Grotthuss mechanism [132].

In addition to cardiolipin, phosphatidylethanolamine is another non-bilayer lipid found in the mitochondrial membrane [133]. Cardiolipin, a phospholipid found exclusively in mitochondrial membranes [134,135], is unequally distributed between the outer (OMMs) and inner mitochondrial membranes (IMMs). Cardiolipin constitutes about 3% of lipids in OMMs and approximately 25% in IMMs [135]. In contrast, phosphatidylethanolamine constitutes about 40% of lipids in both OMMs and IMMs [134,135] and is also found in the plasma membrane and membranes of numerous cell organelles. Being non-bilayer lipids, both phospholipids generate invaginations in tubular cristae and other membrane folds in the IMM via exerting asymmetrical lateral pressure in areas of lipid alkyl chains and lipid polar heads, causing mechanical stress that is relieved through formation of membrane folds [136]. It should be worth noting that the contributions of cardiolipin and phosphatidylethanolamine to the overall structural stability of the IMM and the functional activities of ETC and ATP synthase are different [137]. Cardiolipin, which contains a glycerol bridge between two phosphate groups and four alkyl chains, is a unique molecule characterized by a conical shape and a wide and flexible hydrophobic area with four tails. Hence, cardiolipin possesses the perfect molecular structure for generating tight but flexible bonds between the proteins of respiratory chain super-complexes and ATP synthase in dimeric and oligomeric forms [15,20,26,137,138,139]. Importantly, ATP synthase dimers glued together with cardiolipin facilitate the formation of the apex of the cristae [15]. Experiments performed with giant unilamellar vesicles showed that cristae-like morphology was observed only in lipid vesicles containing cardiolipin and not in other phospholipids [136]. In highly respiring mitochondria, the cardiolipin mediates the oligomerization of ATP synthases and the arrangement of respiratory complexes into super-complexes to promote coupling and enhance OxPhos and the efficiency of ATP synthesis in the mitochondria [15,21,53]. In resting mitochondria, the oligomers of ATP synthases and the super-complexes of respiratory chains consist of smaller complexes (e.g., dimers) that change into a macroscopic cristae-like shape morphology [21,53]. Experiments performed in cardiolipin-containing membranes have suggested that the “breakdown” or “simplification” of oligomers of ATP synthases and respiratory chain super-complexes into smaller units is regulated via the localized proton flow [136]. In addition, it has been shown that cardiolipin in mitochondria plays an important role in regulating changes of the morphology and dynamics of the IMM under pH modulation caused by localized proton flow [137]. It is conceivable that deprotonation and protonation of cardiolipin’s phosphate groups may underlie transitions associated with oligomerization and de-oligomerization of ATP synthases as well as the formation and breakdown of respiratory chain super-complexes.

Additional studies have been performed to shed light on the mechanism of transition from non-bilayers to bilayers in mitochondria. Indeed, through employing differential scanning microcalorimetry and fluorescence emission polarization assays, our research group showed that cardiolipin significantly lowered the energy of phase transition from lipid-ordered to lipid-disordered phase separation (bilayer to non-bilayer phase transition) [129]. The bilayer to non-bilayer transition in cardiolipin containing large unilamellar liposomes did not compromise the permeability of the membrane to potassium ferricyanide [129]. The energy of transition from the bilayer to non-bilayer phases in liposomes with phosphatidylethanolamine was higher than that in liposomes with cardiolipin, whereas the permeability of liposomes to potassium ferricyanide during phase transition with phosphatidylethanolamine was compromised [129]. For more details regarding the mechanisms of lipid-ordered and lipid-disordered phase transitions, the readers are encouraged to read the research article by Puff’s group [140]. Overall, based on the aforementioned studies, we can conclude that one of the main roles of cardiolipin relates to maintenance of the plasticity and structural integrity of the curved areas of the IMM via preserving elastic bonds between ATP synthase dimers and oligomers and between proteins of “respiratosomes”, while the primary role of phosphatidylethanolamine is to supplement cardiolipin in maintaining plasticity of the IMM [15,137].

The ability of cobra venom cytotoxins to induce non-bilayer structures in model membranes enriched with acidic phospholipids was established for the first time in 1986 by our research group, evidenced in electron paramagnetic resonance studies carried out in multilamellar lipid films containing doxyl-stearic acids [141]. Later studies revealed that cobra venom cytotoxins had high binding affinity to cardiolipin [2,4,15,18,20,21,22,26,28,36,37]. In native mitochondria, additional physico–chemical studies have corroborated the affinity of cytotoxins to cardiolipin. Indeed, CTI and CTII from *Naja naja oxiana* and cardiotoxin VII4 from *Naja mossambica mossambica* induced non-bilayer structures via specifically targeting cardiolipin in mitochondrial membranes [2,20,36,37]. In mouse primary cortical neurons and in human SH-SY5Y neuroblastoma cells, our research group showed that cytotoxins can translocate to mitochondria to induce aberrant mitochondrial fragmentation and swelling, a decline in oxidative phosphorylation, and a decrease in ATP synthesis, causing cell lysis [36]. In model membranes that mimic the phospholipid composition of the IMM, cytotoxins increased membrane permeability and triggered the formation of non-bilayer structures, pathological events that preceded apoptosis and necrotic cell death [37]. Interestingly, a very small concentration of highly purified CTI or CTII was sufficient to trigger the formation of non-bilayer signals in isolated, intact mitochondria, as observed via ^31^P-NMR spectroscopy showing the presence of a signal at 6 ppm and 0 ppm, along with a concomitant increase in mitochondrial ATP production [20,21,26,131]. The ^31^P-NMR signals at 6 ppm and 0 ppm were previously observed in cardiolipin containing liposomal membranes treated with cytotoxins CTII and CTI [128,129]. Application of DANTE saturation pulses via ^31^P-NMR spectroscopy revealed that phospholipids responsible for the signal at 6 ppm did not exchange with phospholipids of the lamellar phase, while phospholipids responsible for the signal at 0 ppm exchanged freely with phospholipids of the lamellar phase [20,26,131]. These studies also showed that the signal at 6 ppm originated from non-bilayer-arranged cardiolipins tightly bound to lysine and arginine residues on molecular surface of the cytotoxins [129]. As mentioned previously, cytotoxins CTII and CTI phenocopy the membranotropic activity of the C8 subunit of the F_o_ of bovine ATP synthase [20,26,131]. Similar observations were seen in multilamellar liposomes that mimicked the phospholipid composition of IMM and had either reconstituted cardiotoxin CTII or the C8 subunit of the F_o_ of bovine ATP synthase. Indeed, ^31^P-NMR spectra of these two types of multilamellar liposomes with either cytotoxin CTII or the C8 subunit showed virtually identical results, with non-bilayer signals at 6 ppm and 0 ppm [20,26,131]. Similar results were obtained in multilamellar liposomes with a phospholipid composition of IMM with reconstituted cytotoxin CTI [37]. The phospholipids responsible for the signal at 6 ppm were called immobilized non-bilayer phospholipids, while phospholipids responsible for the signal at 0 ppm were termed mobile non-bilayer phospholipids [20,21]. Immobilized phospholipids responsible for the ^31^P-NMR signal at 6 ppm were also recorded in untreated mitochondrial samples in a temperature range from 15 to 40 °C. In that study, the percentage of immobilized phospholipids increased with the increase in temperature [20]. It should be noted that post-treatment with phospholipase A_2_ (PLA_2_) of mitochondrial samples that had initially been treated with cytotoxin CTII significantly reduced the percentage of immobilized phospholipids. Overall, this breakthrough observation suggested that the increase in percentage of immobilized phospholipids was directly proportional to increase in mitochondrial ATP synthesis [20,21,37] (Figure 5).

Figure 5 shows and other research groups have published experimental data acquired from native mitochondria and treated with cytotoxins CTI and CTII. Model membranes of IMM with the C8 subunit of the F_o_ of ATP synthase and cytotoxins CTI and CTII suggest a new role for cardiolipin. Briefly, this conceptual model proposes that CTI and CTII interact with cardiolipin and subunits of F_o_ to trigger the formation of non-bilayer structures, which alters the activity of ATP synthase. Consistent with this view, the excellent review by Kocherginsky [42] describes many roles of cardiolipin including regulating the activities of ETC proteins, remodeling the IMM, dimerization of the ATP synthase, the transport of protons, and other functions. However, while that review does not describe how cardiolipin modulates ATP synthase activity via formation of non-bilayer structures, here, we report the molecular mechanism through which cardiolipin regulates ATP synthase, based on prior work with cytotoxins. Briefly, cardiolipin transports protons to the ATP synthase along the surface of the IMM facing the intermembrane space [42,49]. The nano-level hydrodynamic effects caused by the proton’s movement along subunit of the F_o_ of ATP synthase induce the spinning of the rotor which then triggers release of ATP from the F_1_ surface into the solution in the matrix [42]. Cardiolipin also binds to the conserved lysine residues K43 at the exit and K7 at the entrance of the ATP synthase rotor [142], which also affect the spinning of the rotor [138]. Cardiolipin, which glues two ATP synthases into dimers, increases the surface curvature of the inner membranes of cristae not only at the apex of the cristae, but also at the peak of the rafts, with each ATP synthase of the dimer siting on the opposite sides of the raft’s peak [15,143]. An increase in membrane curvature at the apex and in the peaks of rafts triggers redistribution of electric charge, pushing protons into the area within the maximum surface curvature, which is where the ATP synthase dimers are located [15,143]. This physiological effect increases the density of protons at the F_o_ unit of ATP synthase to increase the rate of ATP production [21].

Another way for the cristae to increase the rate of ATP synthesis is via the compartmentalization of cristae driven by the bilayer-to-non-bilayer transitions in the IMM [20]. The formation of compartments containing ATP synthase dimers decreases the volume of inter-cristae space, which increases the concentration of protons in bulk water to promote ATP synthesis [20,139]. The compartmentalization hypothesis was upgraded in 2021, stating that protons are not delocalized in bulk water but are localized on the membrane surface to concentrate more protons at the F_o_ units of ATP synthase dimers [53]. Crista compartments are formed through the generation of an intermembrane junction in the intra-cristae space, specifically between areas of maximum curvatures protruding from the rafts on opposite membranes of the cristae [15] (Figure 6).

The initial contact between the rafts on opposite membranes of cristae is driven by the attraction of cardiolipin on the inner surface of the raft to a cationic protein on the inner surface of the opposite raft. The molecular mechanism relating to the formation of the intermembrane junction, which is similar to the schematic 5 in Figure 1, has been described previously [20,21,37,53]. A cationic protein, such as cobra venom cytotoxin, becomes the point of intermembrane contact, which facilitates the bilayer-to-non-bilayer transitions that generate an inverted micelle which then transforms into an intermembrane junction in the intra-cristae space between opposite membranes of the cristae [131]. Cationic proteins that are naturally found in cristae and intra-cristae space, which can also trigger formation of intermembrane junctions, include Cyt *c*, creatine kinase, or misfolded subunits of *c*-rings not incorporated into ATP synthase *c*-rings [21]. Intermembrane junctions are transient structures that are formed and disappear in response to changes in proton density at the cristae’s inner membrane surface [21]. Due to the transient nature of intermembrane junctions, they do not interfere with the transport of biomolecules in the bulk solution found in the intra-crista space [21].

A recent study demonstrated that cardiolipin on the matrix side of the crista leaflet binds selectively to the conserved lysine K43 residue of the ATP synthase rotor [142]. Cardiolipin on the leaflet at the side of the cristae’s intra-cristae space binds selectively to the conserved lysine K7 residue of the ATP synthase rotor [138,142] (Figure 7) to “lock” the rotor to stop ATP synthesis [138]. However, the residence time of cardiolipin bound to the rotor is quite short and is not sufficient to block the required rotation of the c-ring [138]. This is due to protons located in the intra-cristae space of the rotor, which neutralize phosphate groups of cardiolipin bound to K7 [4,15], whereas protons transferred through the F_o_ to the upper leaflet of crista [138] neutralize phosphate groups of cardiolipin bound to K43 [4,15]. Neutralization of phosphate groups removes cardiolipin from the conserved lysine residues, which unblocks the spinning of the rotor of ATP synthase and resumes the ATP synthesis. Neutralization of phosphate groups also increases the conical shape of cardiolipin, which promotes the formation of inverted micelles that entrap protons from the F_o_ subunit into the inner volume of the micelle to facilitate the transport of protons to the proton pumps of respiratory complexes [4,15] (Figure 8B). It is worth noting that the size of the inverted micelles transporting protons is quite small and the movement of inverted micelles does not compromise the barrier function of the membrane as it keeps the membrane impermeable to [Fe(CN)_6_]^3−^ ions, as previously described [37,145]. Thus, inverted micelles serve as nanocarriers of protons through the low dielectric environment of membrane alkyl chains to the proton pumps of respiratory complexes. Specifically, ATP synthesis can be blocked when other deprotonated cardiolipin molecules bind to conserved lysine residues on the rotor. As mentioned above, this happens for a short time as a constant flow of protons moving next to the F_o_ unit “unblocks” ATP synthesis via removing cardiolipin from the conserved lysine residues to promote formation of another inverted micelle (Figure 8C), which transports protons to the proton pumps of respiratory complexes.

How do cytotoxins facilitate ATP synthase via induction of non-bilayer structures in the cristae? Cytotoxins rich in lysine residues target cardiolipin bound to the conserved lysine residues of the F_o_ subunit. As cytotoxins have higher binding affinity to cardiolipin relative to the subunits of the F_o_ [20,37,131], cytotoxins detach cardiolipin from the subunits of the F_o_ unit. This unblocks the ATP synthase rotor and promotes production of ATP. Cytotoxin, which detaches cardiolipin from the F_o_ rotor, settles in the inner membrane surface of the cristae and the cationic polar core head of cardiotoxin can attract phosphate groups of cardiolipin on the opposite inner membrane surface of cristae. Thus, cytotoxin serves as the initial intermembrane contact point to promote the intermembrane junction and formation of compartments in the cristae. In addition, cytotoxin settled in the inner membranes of cristae may perturb the membranes to promote bilayer-to-non-bilayer transition leading to the formation of inverted micelles without compromising the structural integrity of the cristae membranes, through a mechanism that highlights vesicle-to-micelle transition t sub-solubilizing detergent concentrations [146]. The non-bilayer ^1^H-NMR signal in cardiolipin-containing liposomes derives from small inverted micelles in the liposomal membrane which are not permeable to [Fe(CN)_6_]^3−^ ions [37,145]. These small inverted micelles may serve as nanocarriers of protons, as originally proposed by us, for promoting the translocation of protons from the intra-cristae space to the matrix [4,15]. However, they may also serve in transporting protons inside the membrane to the proton pumps of respiratory complexes. Thus, a small concentration of highly purified cytotoxins may “rejuvenate” mitochondrial bioenergetics (increasing ATP synthesis and transmembrane potential) through facilitating compartmentalization, which agrees with the ^31^P-NMR signal at 6 ppm responsible for immobilized phospholipids that do not exchange with the lamellar phase. The data also suggest the formation of small inverted micelles, as evidenced by the presence of the ^31^P-NMR signal at 0 ppm deriving from mobile phospholipids that freely exchange with the lamellar phase and concurring with the ^1^H-NMR signal associated with non-bilayers. It should be noted that the small inverted micelles may suggest the existence of proton currents inside the coupled membrane, as depicted in Figure 3 and proposed by Morelli et al. [40,93].

## 4. Neurodegenerative Pathologies and Cytotoxins

Amyloidogenic proteins and cytotoxins: Proteins that are enriched with β-strands have a high propensity to aggregate into amyloid fibrils via strand–strand interactions. The deposition of aggregated amyloid fibrils into intracellular inclusions and plaques is associated with various brain pathologies including dialysis-related amyloidosis, type II diabetes, Parkinson’s disease, and Alzheimer‘s disease, which develop with aging, particularly in human subjects over 65 years of age [147,148]. An example of proteins that can aggregate includes the amyloidogenic regions found in proteins such as amyloid-β_1–42_, (Aβ42), tau, and serum amyloid A1. These amyloidogenic proteins aggregate into amyloid fibrils and contribute to neurodegeneration of affected neurons in these neurodegenerative diseases [149,150,151,152,153,154]. Pharmacological interventions to reduce amyloidogenic proteins have not had success in clinical trials in reducing clinical symptoms of disease, presumably due to the low efficiency of large antibodies in disaggregating protein aggregates in vivo [155,156,157]. However, one of the novel strategies for developing an efficient therapeutic drug against amyloidosis is based on designing amyloidogenic sequence “traps” that stop the transformation of amyloidogenic proteins into amyloid fibrils (Figure 9) [148]. The customized amyloidogenic sequence traps feature anti-parallel β-sheets, a structure that is analogous to that found in cobra cytotoxins. While thought-provoking and speculative, the similarity in secondary structures between amyloidogenic traps and cytotoxins, their small molecular size, and similar biophysical characteristics give some credence to the concept that cytotoxins can act as amyloidogenic traps and warrants future investigations to provide definitive experimental evidence through in vitro or in silico assays to test their ability to bind to amyloid-like proteins (α-synuclein and β-amyloid).

A variety of amyloid-like proteins interact with membranes of organelles and plasma membranes to form aggregates, which lead to the degeneration of neurons in various neurodegenerative diseases [36]. Neuronal dysfunction and degeneration in Parkinson’s disease are associated with abnormal aberrant folding and aggregation of the α-synuclein (α-S) protein in affected areas of the brain [158,159]. α-S in conjunction with ubiquitinated proteins forms the core of large intracellular inclusions termed Lewy bodies in Parkinson’s disease, which contribute to the neurodegeneration of dopamine and cortical neurons. When highly phosphorylated, α-S is highly pathogenic in Parkinson’s disease as it facilitates the assembly of aggregates at the presynaptic plasma membrane, which disrupts neurotransmission [160,161]. The oligomerization of α-S proteins at the presynaptic membrane is a hallmark feature in the neuropathology of Parkinson’s disease [162,163]. Parkinson’s disease pathogenesis is associated with disruption of cell membranes [164] including mitochondrial membranes via oligomerization of α-S [165] leading to disruption of membrane structure, which increases the membrane permeability of mitochondria and the formation of membrane channels and pores [163]. It should be noted that targeting of cardiolipin via α-S may be a starting point for pathological development associated with Parkinson’s disease [165].

The effects of binding of α-S to mitochondrial membranes have been a central focus of several research studies. Like cytotoxins at high concentrations, the intracellular over-expression of α-S in primary neurons and dopaminergic cell lines is highly pathogenic as it leads to mitochondrial pathology including loss of mitochondrial membrane potential, mitochondrial swelling and fission, the subsequent release of cytochrome c, an increase in ROS levels (e.g., superoxide), an increase in mitochondrial Ca^2+^, bioenergetic failure, and the onset of apoptosis [166,167,168,169]. Like monomeric α-S, treatment of neuronal cells with oligomeric α-S triggers similar pathological effects but with higher potency [167,169,170]. These effects have been explained through the binding of α-S oligomers to cardiolipin in the IMM causing perturbation of the IMM structure [165]. It is noteworthy that the loss of activity of the mitochondrial ADP/ATP carrier, a membrane-bound protein stabilized by cardiolipin, has also been linked to the binding of oligomeric α-Sb to cardiolipin [171,172]. The formation of amyloid or lipidic pores in mitochondrial membranes is also caused by the binding of oligomeric α-S to cardiolipin [173]. The aggregated forms of other amyloidogenic proteins such as amyloid beta and tau detrimentally affect the structural integrity of mitochondrial membranes in similar ways to α-Sb [174].

Cobra venom cytotoxins share many physiological activities with amyloidogenic proteins, such as formation of transmembrane pores via the oligomerization of cytotoxins [27], disruption of membrane structure via altering the lipid packing state of lipid bilayers [2], increase in membrane permeability [24], and targeting of cardiolipin leading to aberrant alterations in the structure of the IMM [2,20,26,36,37,131]. Cobra cytotoxins also share structural features with amyloidogenic proteins, such as including a central region that contains a high concentration of hydrophobic residues that have a high propensity for adopting a β-sheet structure, harbor acidic residue(s) located in the C-terminal domain, and contain a positively charged N-terminal region [1,2]. However, it is worth noting that there have been very few studies on whether cytotoxins can be used as pharmacological tools to understand the functional mechanisms of nerve cells in health and pathology [1,2]. Although few studies have been conducted, some of them that have stemmed from our research group have allowed us to gain a better understanding of the therapeutic potential of cytotoxins in age-associated neurodegenerative diseases via restoring mitochondrial bioenergetics [36]. While cobra cytotoxins are not widely known for their neurotoxicity, several research reports have shown that cytotoxins can interact with cells of the nervous system. The pathophysiological effects of cytotoxins are based on the ability of cytotoxins to exert neurotoxicity through promoting axonal degeneration and disrupting neurotransmission via the deregulation of the activity of cell membrane-bound enzymes and receptors and directly depolarizing excitable membranes of neurons [1].

In the next subsections, we focus on our previous work in which we showed that cobra venom cytotoxin VII4 from Naja mossambica mossambica, similarly to amyloidogenic proteins, promoted neurotoxicity through translocating mitochondria to disrupt mitochondrial membrane structure and function via a mechanism that involved targeting cardiolipin and triggering bilayer-to-non-bilayer transition in mitochondrial membranes.

Mechanisms of mitochondrial targeting: implications of cytotoxin-induced apoptosis in neuronal systems: While the mechanisms through which cytotoxins can bind to mitochondria are beginning to be elucidated in various cell types, it is less clear how cytotoxins can trigger mitochondrial dysfunction and activate cell death pathways. Most of the information regarding how cytotoxins promote mitochondrial dysfunction have been carried out in neuronal systems. In a recent study, we examined the interaction of cytotoxin VII4 from Naja mossambica mossambica with mouse primary cortical neurons and human SH-SY5Y neuroblastoma cells. Briefly, in SH-SY5Y cells treated with fluorescently labeled cardiotoxin VII4, confocal microscopy studies showed that cytotoxin could rapidly translocate from the cell membrane to mitochondria and colocalize with mitochondria to promote aberrant mitochondrial fragmentation, a decline in oxidative phosphorylation, and a decrease in energy production [36]. Consistent with our study, another study performed in SKN-SH neuroblastoma cells showed that cytotoxins from the Taiwan cobra rapidly colocalized to mitochondria as an index of translocation to induce significant mitochondrial depolarization, which was accompanied by an increase in reactive oxygen species, pathological conditions that led to apoptosis [35]. Based on these studies [2,35] and our results with cytotoxin VII4 on cortical neurons and SH-SY5Y neuroblastoma cells [36], we proposed that cytotoxins were able to interact with cardiolipin in mitochondrial membranes. To test our hypothesis that cytotoxins can interact with the IMM and to corroborate the aforementioned findings that cytotoxins can colocalize with mitochondria, we employed biophysical methods, phosphorescence quenching assays, electron paramagnetic resonance spectroscopy, and membrane permeability assays via ^1^H-NMR to study the interaction of cytotoxin VII4 with liposomes that mimicked the composition of the OMM and the IMM. This study was complemented with computer simulations: molecular dynamics (MD) and AutoDock ligand–receptor interactions.

For phosphorescence quenching assays, we employed erythrosine phosphorescence quenching with ferrocene. In this system using a phosphorescence probe (erythrosine) and its quencher (ferrocene), the lifetime of phosphorescence depended on the viscosity of the hydrophobic area of the membrane as in this pair of probe and quencher, both nonpolar molecules diffused in the hydrophobic area [2,175,176]. Treatment of pure phosphatidylcholine-containing liposomes with cytotoxin VII4 did not affect the phosphorescence lifetime, proving that cytotoxin did not interact with membrane made of phosphatidylcholine. However, treatment of liposomes containing either 5% or 20% cardiolipin with cytotoxin VII4 significantly increased the lifetime of phosphorescence, and this effect was more pronounced in liposomes with 20% cardiolipin [36], which strongly suggested that cytotoxin VII4 targets cardiolipin.

To investigate membrane permeability and the structural polymorphism of liposomal membranes treated with cytotoxin VII4, we used ^1^H-NMR spectroscopy with paramagnetic ferricyanide ions [Fe(CN)6]^3−^. As expected, cytotoxin VII4 did not affect the ^1^H-NMR spectrum of pure phosphatidylcholine-containing liposomes. However, in liposomes with cardiolipin, cytotoxin VII4 induced the formation of non-bilayer phases and increased the permeability of the membrane to [Fe(CN)6]^3−^ ions. These effects were more visible in liposomes containing 20% cardiolipin. Noteworthily, small concentrations of cytotoxin VII4 did induce formation of the non-bilayer phase without compromising membrane structural integrity since the membranes were impermeable to [Fe(CN)6]^3−^ ions in liposomes with 5% and 20% cardiolipin [36].

To further investigate the formation of the non-bilayer phase induced via cytotoxin VII4 in cardiolipin-containing membranes, we employed electron paramagnetic resonance (EPR) using a 5-doxyl stearic acid spin probe (5-DSA) in oriented multi-bilayer lipid films of pure phosphatidylcholine and lipid films containing either 5% or 20% cardiolipin. A strong anisotropy of the EPR spectra, that is, when spectral lines did not coincide at different angles of the lipid film in the applied magnetic field, indicated the existence of a highly ordered lipid bilayer (Figure 10A at 0 cytotoxin–lipid molar ratio). EPR spectral isotropy when spectral lines coincided at different angles in the lipid film in the applied magnetic field indicated a highly disordered lipid packing due to the formation of a non-bilayer phase (Figure 10B at 0.04 cytotoxin–lipid molar ratio). Treatment of multibilayer lipid films with cytotoxin VII4 did not affect the EPR spectral lines of pure phosphatidylcholine lipid films but did affect the spectral lines of lipid films containing cardiolipin, and spectral isotropy was more pronounced in lipid films with 20% cardiolipin (Figure 10A,B at 0.04 cytotoxin–lipid molar ratio). Quantitative analysis of the EPR spectra of 5-DSA in multibilayer oriented lipid films treated with cytotoxin VII4 revealed the existence of a non-bilayer phase with immobilized lipids, which we presumed was cardiolipin directly interacting with cytotoxin VII4 [36].

Molecular Dynamics and AutoDock simulation of the interaction of cardiolipin with cytotoxin *VII4:* To confirm that cytotoxin VII4 interacted with cardiolipin, we performed molecular dynamics simulations using the Gromos53a6 force field [177]. In these computer simulations, the system was set up to study how cytotoxin VII4 was predicted to interact with a phospholipid bilayer made of only phosphatidylcholine or with a phospholipid bilayer made of phosphatidylcholine and cardiolipin having a phospholipid composition similar to the OMM. In the two molecular dynamics (MD) simulations (20 ns each), cytotoxin VII4 failed to reach the inter-atomic threshold of interaction with a virtual membrane made of only phosphatidylcholine. However, in a virtual bilayer made of phosphatidylcholine and cardiolipin, two MD runs showed that cytotoxin VII4 rapidly bound with one cardiolipin molecule and subsequently interacted with a second cardiolipin molecule (Figure 11B). Similar results were obtained when running the same MD experiments with cytotoxins of different cobra species, including cytotoxin CTX1 and cytotoxin II from the venom of cobra *Naja atra*. Overall, our MD results strongly suggest that cytotoxin VII4 and possibly most other cytotoxins target cardiolipin molecules in mitochondrial membranes via electrostatic attraction by means of ionic and hydrogen bonding [36]. However, while our simulation studies give credence to the notion that cytotoxins can target mitochondria via binding to CL and other anionic phospholipids, the ability of CTX1 and other cytotoxins to penetrate the mitochondria and disrupt bioenergetics depends on the number of cytotoxin molecules (concentration) that can interact with cardiolipin on the OMM, the type of cytotoxins (CTX1 vs. cardiotoxin 2), the extent of translocation to the IMM, and formation of non-bilayer structures, which can lead to toroidal structures.

To identify potential cardiolipin-binding sites on the molecular surface of cytotoxin VII4, we simulated molecular docking of cytotoxin with cardiolipin employing the AutoDock Vina Version 4.2 program [178]. To study the long-range interactions of cardiolipin with cytotoxin in aqueous solution, we used only the polar head of cardiolipin and truncated the alkyl chains of the cardiolipin. The rationale for this is that the alkyl chains are confined within the hydrophobic area of the bilayer and therefore cannot be involved in interaction with cardiotoxin in water. To simulate the interaction of cytotoxin when cytotoxin was already submerged into the membrane, we used the complete molecule of cardiolipin with four alkyl chains. For comparison, we also simulated docking of cytotoxin with the polar head of phosphatidylcholine and with the complete phosphatidylcholine molecule. For each docking pair, the molecular docking runs produced nine top-ranked phospholipid–cytotoxin binding structures, based on their enthalpy energy released upon the binding of phospholipid with cytotoxin. In these computer simulations, we observed that the interactions of cytotoxin with either the polar heads or the complete molecular structure of the lipids involved ionic, ion-polar, and hydrogen forces of attraction.

It should be noted that in all docked pairs, the interaction of cytotoxin with the polar heads or the complete molecule of cardiolipin was predicted to release more enthalpy energy than when simulations with phosphatidylcholine were conducted. These data suggest that cardiolipin is the phospholipid that cytotoxin targets in mitochondria [36]. It is noteworthy that all docked structures contained four basic residues, K12, K18, K35, and R36, on the molecular surface of the cytotoxin that interacted with cardiolipin. Interestingly, all four basic residues were conserved for all cobra venom cytotoxins [1]. All docked structures of cardiotoxin bound to the complete phospholipid molecules were predicted to release more enthalpy energy than structures of cytotoxin docked with the polar heads of phospholipids. This observation suggests that the alkyl chains of lipids are important in mediating the interaction of cell membranes with cytotoxin [36]. Interestingly, in all docked pairs of cytotoxin with phosphatidylcholine, the basic choline group was not predicted to interact with acidic residues of cytotoxin, E16 and D57 [36]. This observation suggests that the physiological role of phosphatidylcholine may be to serve as a barrier, repelling basic proteins even with acidic residues.

Overall, the data provided by the MD and docking studies suggest that cytotoxins target cardiolipin in mitochondria to promote its translocation and affect ATP synthase activity. In addition, these MD simulations suggested that the initial interaction of cytotoxin with the mitochondrial membrane involves long-range ionic interactions, whereas London dispersion forces mediate the interaction of cytotoxin with the membrane when cytotoxin penetrates the lipid bilayer.

The consequences of cytotoxins targeting mitochondria at high concentrations can have detrimental effects on cells via activating apoptosis. However, aside from apoptosis, cytotoxins have been well characterized to activate cell cycle arrest, necrosis, and necroptosis in some instances [179]. Importantly, the signaling pathways and cell death effectors activated by cytotoxins are beginning to be elucidated. Indeed, a majority of studies suggest that cytotoxins can activate apoptosis via activating the Ca^+2^/protein phosphatase 2A/5′ adenosine monophosphate activated protein kinase signaling pathway, increasing the level of ROS, releasing apoptogenic factors such as cytochrome C, or activating the p39 MKAP/c-Jun-mediated signaling pathway. In addition, cytotoxins can have the capacity to affect the extrinsic pathway of apoptosis through activating cell death receptors via Fas/FasL [180]. A myriad of tissues start the extrinsic pathway of apoptosis through activating cell death receptors like tumor necrosis factor. Beyond the mitochondrion, other studies have shown that cytotoxins can activate necrosis through translocating to the lysosome to disrupt its membrane integrity and cause the release of Cathepsin B. The extent of lysosomal disruption and the amount of cathepsin B are correlated to the extent that necrosis is activated via cytotoxin-mediated toxicity vs. apoptosis [180]. However, the specific physico–chemical conditions and factors that lead to activation of apoptosis vs. necrosis are yet to be identified in different cell systems, including cancer cells [180,181]. It is conceivable that the extent of mitochondrial targeting, concentration (absolute accumulation in mitochondria), affinity to cardiolipin, and the extent of the formation of non-bilayer structures are factors that can contribute to mitochondrial dysfunction and activation of apoptosis. Unlike snake venom neurotoxins, which have a high specificity to neuronal tissues via binding to cell membrane receptors, it is clear that many of the degenerative effects of cytotoxins on neurons are passive, mainly due to the ability to bind to anionic phospholipids that reside in myelin sheaths and translocate neuronal cell membranes. There have been a few studies that have analyzed the mechanisms through which cytotoxins can induce neurodegeneration in neuroblastoma cells and primary cortical neurons. However, some questions remain that warrant future studies to analyze whether cytotoxins can pass through the blood–brain barrier and whether there is preferential targeting of cytotoxins towards myelinated vs. unmyelinated neurons and other cell types in the brain, such as oligodendrocytes or glia. Therefore, future studies are warranted to functionally dissect the conditions that lead to apoptosis vs. necrosis.

## 5. Conclusions

In this review, we have briefly outlined the basics of chemiosmotic theory proposed by Peter Mitchell [39]. We mainly focus on the controversial aspects which have been the subject of intense investigation. The main feature of this review is its provision of a compressive understanding of the role of non-bilayer structures in energy-transducing membranes, an area of bioenergetics which has been meticulously addressed by Dr. Gyozo Garab’s research group [21] and by our research group [20]. Out studies suggest that cobra cytotoxins at high concentrations target the mitochondria to disrupt the structure/function of the mitochondria, presumably due to the formation of non-bilayer structures that form toroidal pores that disrupt the structural integrity of mitochondrial membranes [34,35,36,37]. However, at low concentrations, cytotoxins can increase ATP synthase activity in fully functional mitochondria via inducing the formation of two types of non-bilayer structures: one with immobilized phospholipids which do not exchange with the lamellar phase, and another with phospholipids of high-speed isotropic mobility which exchange with the lamellar phase [20,26,131]. These data have changed our view regarding the role of non-bilayer phase in modulating mitochondrial ATP synthesis and advance the field of membrane biology and bioenergetics [4,15,21]. In addition, using a broad range of biophysical and biochemical methods to study how cytotoxins interact with isolated mitochondria or model membranes, we provide the following updated conceptual framework on the role of non-bilayers in the IMM regulating oxidative phosphorylation: (1) the generation of an intermembrane junction in the intra-cristae space leading to compartmentalization in cristae, which increases the concentration of protons along the inner cristae membrane surface leading to maximum proton density to optimize ATP synthase activity [15,20,21]; (2) detachment of cardiolipin from conserved lysine residues on the rotor, which unblocks the spinning of the rotor of ATP synthase to facilitate ATP production [4,15,37]; and (3) the induction of bilayer to non-bilayer structures including the formation of inverted micelles without compromising the structural integrity of the cristae membranes [182]. These inverted micelles transport protons inside the membrane from F_o_ to the proton pumps of respiratory complexes, which ultimately leads to an increase in ATP synthase activity [4,15]. It is worth noting that the above experimental and theoretical findings were garnered thanks to the use of cytotoxins that acted as pharmacological tools to probe the structure and function of the mitochondrial membrane. Therefore, the use of cytotoxins in membrane biophysics and mitochondrial research has greatly accelerated our understanding and shed light on the molecular mechanism of mitochondrial ATP production.

Prior studies that analyzed the interaction of cytotoxin VII4 with mitochondria in mouse primary cortical neurons, human SH-SY5Y neuroblastoma cells, and model cardiolipin-containing membranes [36] suggested that cytotoxin exhibited neurotoxicity via triggering pathological reactions in the mitochondria of nerve cells in a similar manner to amyloidogenic proteins such as alpha-S, betta-amyloid, tau, through targeting cardiolipin. Although cytotoxins do not have an amyloid-like domain, cytotoxins seem to phenocopy important physico–chemical properties of amyloid-like proteins due to the presence of similar electro–chemical and structural properties. These properties include the presence of highly basic charge regions interspersed with hydrophilic and hydrophobic regions, and the presence of lipid-binding domains and anti-parallel β-sheets that enable these proteins to target cardiolipin and penetrate the OMM and IMM to form toroid-like structures. However, future studies are necessary to refine the conceptual model through which cytotoxins promote neurotoxicity (apoptosis vs. necrosis). The groundbreaking observation that cytotoxins can increase the activity of mitochondrial ATP synthase without compromising structural integrity of the mitochondrial membrane at low concentrations may represent a promising pharmacological avenue for rejuvenation of bioenergetics in neurons, which can be used for treatment of neurodegeneration with aging. While this is a technical challenge to overcome in the future, customizing cytotoxins using site-directed mutagenesis (truncations and point mutations) to reduce the “mitotoxicity” of cytotoxins while preserving their ability to increase ATP synthase may be a novel strategy in developing pharmaceuticals for treating neurological disorders. For instance, given that cytotoxin CTI has less propensity to disrupt membranes containing cardiolipin relative to cytotoxin CTII, it would be possible to tailor CTI in a manner that could be conducive to increasing ATP synthase activity. Indeed, the above concepts represent a highly challenging endeavor worth carrying out to develop recombinant cytotoxins that can increase energy production in neurons via increasing ATP synthase activity. We recognize that additional data supporting the therapeutic potential of cytotoxins, including whether they can act as amyloidogenic traps, are hypothetical, as discussed above, but are worth pursuing via engaging in vitro and in silico assays as first steps to acquire proof of concept. Hence, we hope that the molecular insights presented in this review may inspire new strategies for therapeutic intervention.

## Figures and Tables

**Figure 1 toxins-16-00287-f001:**
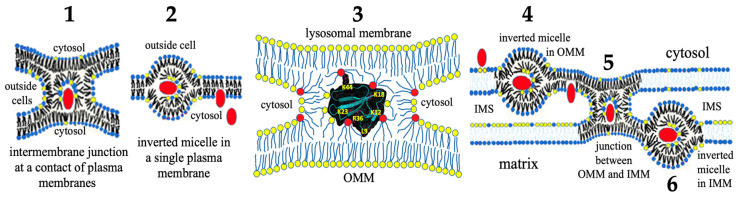
Cytotoxin internalization/translocation through plasma membrane, OMM, and IMM. 1—Intermembrane junction triggered by CTII (red) at a contact of two plasma membranes. 2—CTII settles in an inverted micelle in a single membrane after fission of an intermembrane junction. Due to the high surface curvature, the inverted micelle fuses with the lipid bilayer to release CTII into the cytosol. 3—contact between the outer mitochondrial membrane (OMM) and lysosomal membranes triggered by CTII to generate a new intermembrane junction. Amino acid residues of CTII (illustrated representation), K23, R36, and K12 bind to the OMM surface and L9 penetrates to the hydrophobic region of OMM. Amino acid residues K44, K18, K5, and K35 on another side of the CTII attract acidic lipids of the neighboring lysosome to form an intermembrane junction. 4—CTII settles in the inverted micelle in OMM following fission of the intermembrane junction, which then releases CTII into the intermembrane space (IMS). 5—CTII binds to CLs on the OMM and the IMM to form another intermembrane junction between OMM and IMM. 6—CTII settles in the inverted micelle in IMM after fission of the affected regions of the OMM and IMM. At high concentration, CTII disturbs the integrity of the IMM, whereas CTII at very low concentrations increases the “plasticity” of IMM to facilitate ATP synthesis. In 1, 2, 4, 5, and 6 the head groups of cardiolipins (CLs) are colored yellow and the head groups of phosphatidylcholines (PCs) are colored blue. In 3, the head groups of CLs are colored red and the head groups of PCs are colored yellow. In all schematics, the cytotoxins that permeate and translocate through the membranes are indicated with a large red circle. This figure was modified and readapted from [2,15,37].

**Figure 2 toxins-16-00287-f002:**
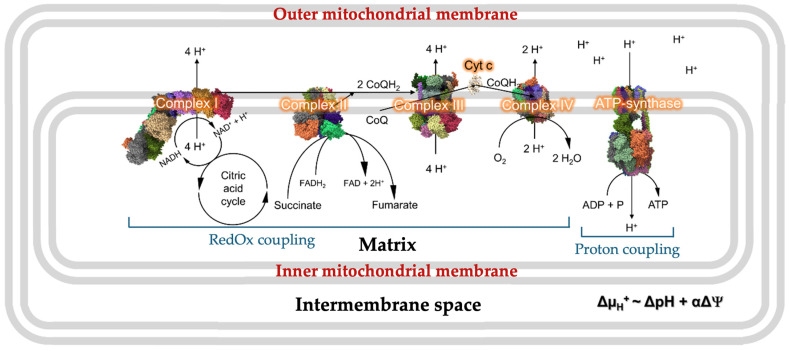
Schematic of the protein complexes of the electron transport chain and ATP synthase embedded in the IMM surrounded by OMM with Δµ_H_^+^ energized states formed between intermembrane space and matrix, utilized for ATP synthesis. The electron and proton transfers are indicated with arrows. The following crystal structures are indicated in the schematic: PDB # 7NYR—Complex I, PDB # 6WU6—Complex II, PDB # 5XTE—Complex III, PDB # 7JRO—Complex IV, PDB # 6C4W—Cyt c, PDB # 6OQW—ATP synthase. Parts of this schematic are modified from [21,40].

**Figure 3 toxins-16-00287-f003:**
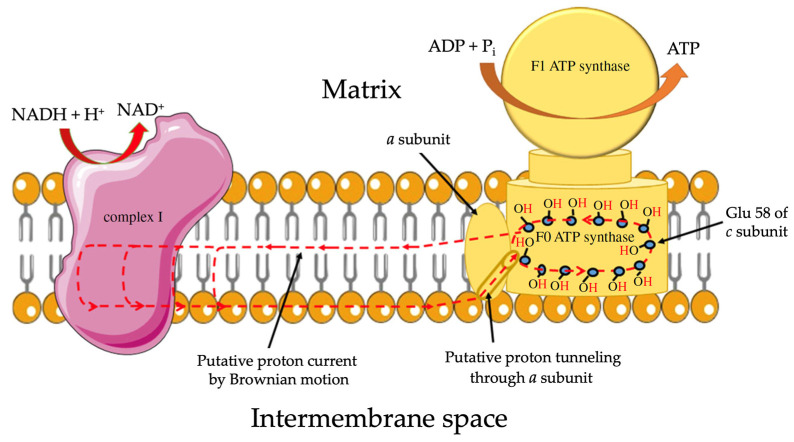
A putative proton circuit inside the IMM. The polar heads of phospholipids on both sides of the IMM are shown as brown spheres. The diagram proposes that the protons (red dotted line) are transferred to the Glu 58 at the center of subunit *c* through subunit *a* of ATP synthase via proton tunneling. Protons flow on the intermembrane side of the IMM surface and are always bound to phospholipid polar heads. This figure has been adapted from [40].

**Figure 4 toxins-16-00287-f004:**
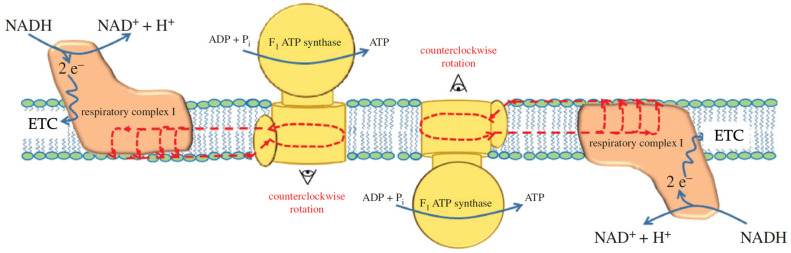
Schematic of a bi-facing ATP synthase arrangement in a myelin sheath, sustaining proton currents delivered via respiratory Complex I. A proton circuit is built entirely inside the myelin membrane. For simplicity, only respiratory Complex I is shown in the schematic. This schematic has been adapted from [93].

**Figure 5 toxins-16-00287-f005:**
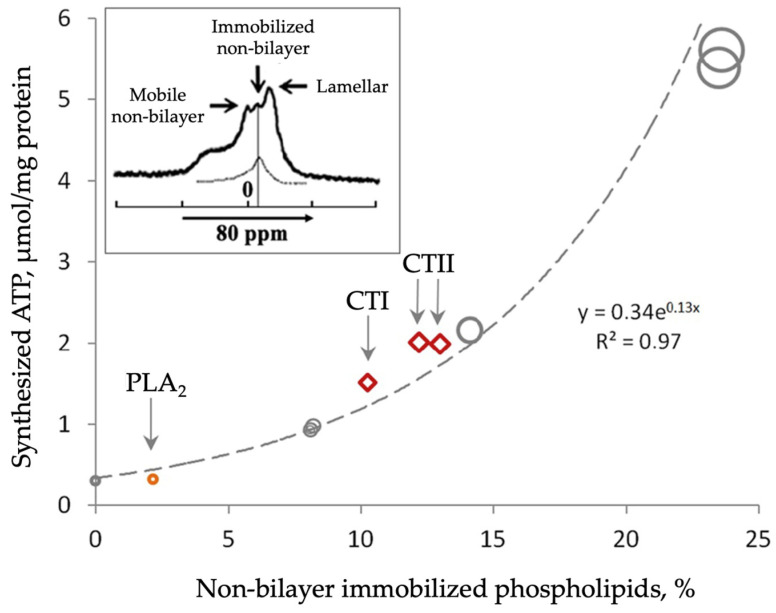
Correlation of ATP synthesis and the percentage of immobilized non-bilayer lipids in mitochondria samples. The percentage of these lipids was varied through increasing the temperature or adding CTI, CTII, or PLA_2_. Immobilized lipids were observed in the ^31^P-NMR spectra at 6 ppm; this signal was retained after applying a DANTE train of saturation pulses to the high-field peak of the lamellar spectral component [20]. The ATP levels, expressed as μmol ATP synthesized per mg of mitochondrial proteins, were monitored via taking measurements from aliquots from the ^31^P-NMR sample tubes. The open circles of progressively increasing size represent measurements taken from untreated mitochondria at 8, 15, 25, and 40 °C. The sizes of the markers overlap the error values. Data points were obtained from [20,37]. Typical ^31^P-NMR spectra are presented in the inset (the thin line below the main spectrum shows the 6 ppm signal remaining after the DANTE train). This figure has been adapted from [21].

**Figure 6 toxins-16-00287-f006:**
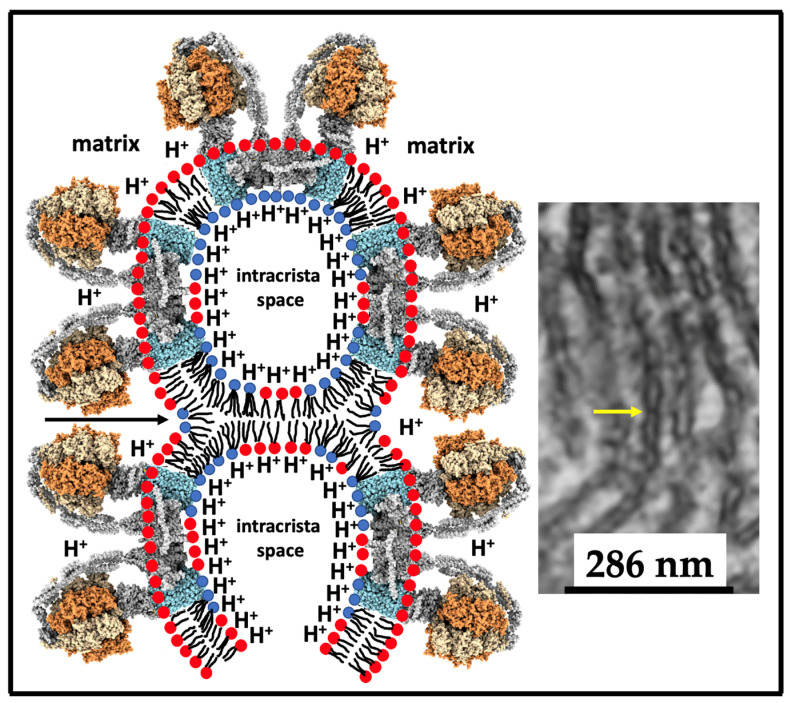
Intermembrane junctions increase the concentration of protons along the crista inner membrane surface. On the left is a schematic of two crista compartments that form rafts on the outer surface of the compartments, which are separated by an intermembrane junction—as indicated with the black arrow. The atomic structure of ATP synthase was reconstituted by Dr. S. V. Nesterov of the Moscow Institute of Physics and Technology (PDB # 6B8H) using the ChimeraX program (UCSF) [144,145]. The α- and β-subunits of the catalytic domain are shown in yellow and orange, respectively; the C ring is shown in blue. All other subunits are shown grey. The polar heads of cardiolipin are shown in blue. Phospholipids with red polar heads represent other phospholipids in IMM with two alkyl chains. To provide better clarity, the alkyl chains of phospholipids are not drawn over the ATP synthase hydrophobic subunits. On the right is a transmission electron microscopy image modified from [53] that shows intermembrane junctions between the membranes of the cristae (yellow arrows) that separate the cristae rafts. This figure is modified from [15].

**Figure 7 toxins-16-00287-f007:**
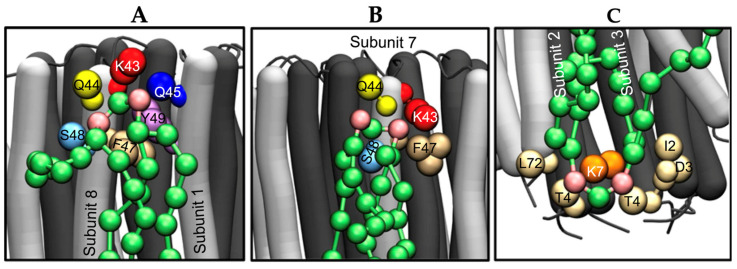
Modes of cardiolipin binding to subunits of C_8_-rings. The N- and C-terminal α-helices of C-subunits are shown as long dark and gray rods, respectively. Cardiolipin phosphate groups are depicted in pink and other groups are shown in green. The large colored spheres representing the coarse-grain beads of specific amino acids lie within 0.7 nm of the phosphate beads of cardiolipin. (**A**,**B**) Schematic showing the head-group region of cardiolipin on the matrix side of the crista leaflet bound to K43 and adjacent residues on C subunits 8 and 1. (**C**) Schematic showing the head-group region of cardiolipin on the intra-cristae side of the crista bound to K7 and adjacent residues of the C subunit. This Figure was adapted from [142].

**Figure 8 toxins-16-00287-f008:**
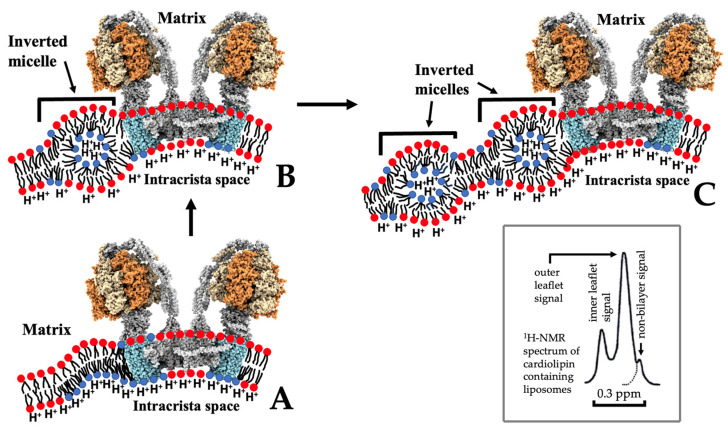
Inverted micelles predominantly made of cardiolipin serve as nanocarriers of protons, transporting protons from the F_o_ unit of ATP synthase to respiratory complexes at the IMM that act as proton pumps. The increased density of protons at the F_o_ unit of ATP synthase increases the conical shape of cardiolipin molecules (**A**) and triggers the formation of inverted micelles carrying H^+^ ions (**B**) inside the membrane to the nearest proton pump. The formation of inverted micelles is coupled to the removal of cardiolipin from the conserved lysine residues of the rotor to promote spinning of the rotor and ATP synthesis in the clockwise direction. Newly formed inverted micelles push previously made inverted micelles (**C**) towards the proton pump. Reconstitution of ATP synthase and designation of phospholipids are described in Figure 6. The inset shows the ^1^H-NMR signal of cardiolipin + phosphatidylcholine liposomes treated with cardiotoxin CTII in solution of [Fe(CN)_6_]^3−^ ions. The non-bilayer signal of inverted micelles coexisting with the intact signals from inner and outer leaflets of liposomal membrane in solution of [Fe(CN)_6_]^3−^ ions suggest that inverted micelles do not compromise the structural integrity of the liposomal membrane. Figures **A** and **B** are modified from [15]. Inset is modified from [37].

**Figure 9 toxins-16-00287-f009:**
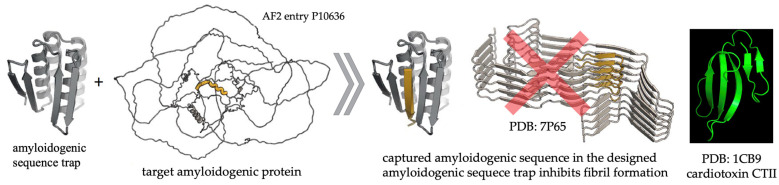
Schematic showing a custom amyloidogenic sequence trap (left, gray) that can bind amyloidogenic sequences (yellow ribbon, right) and block fibril formation. Amyloidogenic sequences form amyloid fibrils through strand–strand interactions (right); AF2, AlphaFold2. Illustrated representation of cobra cardiotoxin CTII with anti-parallel β-sheets, suggesting that cobra cytotoxins could possibly serve as a natural amyloidogenic sequence trap. This figure has been modified from [148].

**Figure 10 toxins-16-00287-f010:**
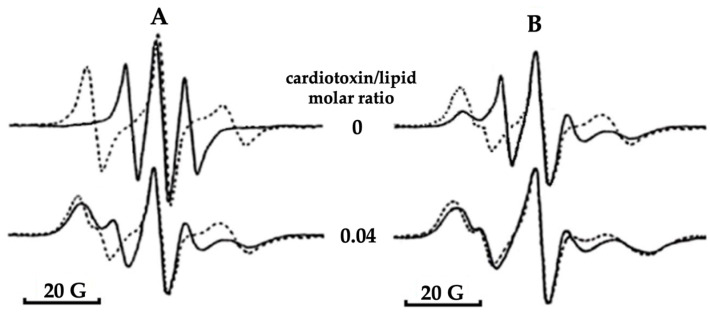
EPR spectra of 5-DSA acid in oriented multibilayer lipid films of phosphatidylcholine + 5% cardiolipin (**A**) and oriented multibilayer lipid films of phosphatidylcholine + 20% cardiolipin (**B**) containing a 5-DSA–lipid molar ratio of 1/100 at 18 °C. The EPR spectra in multibilayer films were recorded with the magnetic field parallel (broken lines) and perpendicular (solid lines) to the bilayer normal. Multibilayer films were treated with cytotoxin VII4 at the indicated cytotoxin-to-lipid molar ratios. This figure has been adapted from [36].

**Figure 11 toxins-16-00287-f011:**
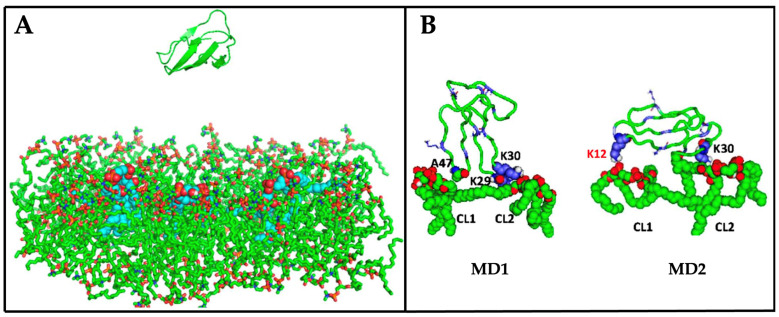
(**A**) The interaction of cytotoxin VII4 (PDB # 1CDT) with an in silico phospholipid bilayer, according to molecular dynamics runs. Cytotoxin, shown as a ribbon of β-strands and loops, was positioned 20 Å away from a bilayer made of 125 molecules of phosphatidylcholine and 3 molecules of cardiolipin. The cardiolipin molecules are shown in cyan, space-filling representation. (**B**) The interaction of cytotoxin VII4 with the same bilayer as in A in two MD runs where phosphatidylcholine molecules were removed to enhance the visibility of cardiolipin molecules. In MD1, the K30 residue of cardiotoxin makes an ionic bond with the phosphate group of one cardiolipin while the A47 residue interacts hydrophobically via hydrogen bonding with another cardiolipin molecule. In MD2, K12 and K30 residues of cytotoxin make ionic bonds with the phosphate groups of two cardiolipin molecules. CL stands for cardiolipin. This figure has been modified from [36].

## Data Availability

The data presented in this study are available in this article.

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
