# Peer review of "Cobra Venom Cytotoxins as a Tool for Probing Mechanisms of Mitochondrial Energetics and Understanding Mitochondrial Membrane Structure"

_toxins, 2024, doi:10.3390/toxins16070287_

Round 1
Reviewer 1 Report
Comments and Suggestions for Authors
A comprehensive review of cobra venom cardio toxins on bioenergetics. A few comments to be made here:
1. I would suggest, instead of cardiotoxins, please use the term cytotoxins. Cytotoxins are the “newer” representative terms for this finger toxin.
2. There have been many reports on the intracellular signaling pathways targeted by cytotoxins, please include those in your review. In addition, there have also been reports on the transition of cell death caused by cytotoxins in concentration-dependent manner I.e., from apoptosis to necrosis/necroptosis (10.1016/j.ijbiomac.2021.06.145 & 10.1016/j.toxicon.2015.09.017), please discuss as well.
3. Internationalisation of cytotoxins and their colocalisation into mitochondria needs to be discussed as well. Cytotoxins have been shown to cause mitochondrial depolarization and this is related to the colocalisation of the toxins on mitochondria.
4. In MD simulations, can authors indicate how long is the simulation run? Does membrane penetration occur? Or it is just merely pore formation? And how can this depolarize the mitochondrial membrane and disrupts bioenergetic?
Comments on the Quality of English LanguageNo big issue with English language, suggest to get proof-reading.
Author Response
Reviewer 1
A comprehensive review of cobra venom cardio toxins on bioenergetics. A few comments to be made here:
- I would suggest, instead of cardiotoxins, please use the term cytotoxins. Cytotoxins are the “newer” representative terms for this finger toxin.
Answer: Thank you for the observation, we agree with the suggestion and have changed the term cardiotoxin for cytotoxin throughout the review manuscript.
- There have been many reports on the intracellular signaling pathways targeted by cytotoxins, please include those in your review. In addition, there have also been reports on the transition of cell death caused by cytotoxins in concentration-dependent manner I.e., from apoptosis to necrosis/necroptosis (1016/j.ijbiomac.2021.06.145 & 10.1016/j.toxicon.2015.09.017), please discuss as well.
Answer: We agree with the aforementioned suggestion and have made appropriate changes to expand on the mechanisms by which cytotoxins activate cell death pathways including apoptosis and necrosis on lines 985-1016 as well as added the suggested references in lines 1471-1476.
- Internalization of cytotoxins and their colocalisation into mitochondria needs to be discussed as well. Cytotoxins have been shown to cause mitochondrial depolarization and this is related to the colocalisation of the toxins on mitochondria.
Answer: We agree with this astute suggestion. We expanded the mechanism by which cytotoxins are internalized and colocalize in mitochondria based on several studies done in neuroblastoma cells and primary neurons in the cytotoxin internalization in lines 852, 853-857. These molecular events and mechanisms of internalization of cytotoxins to the outer and inner mitochondrial membranes are discussed extensively in lines 133-13142 and lines 555 560. In addition, we made additional edits to the figure caption in Figure 1 (line 157) to further highlight the consequences of cytotoxin colocalization involving mitochondrial depolarization, and other events are discussed in the subsection on line 843-847, 851-852 and in lines 853-857.
- In MD simulations, can authors indicate how long is the simulation run? Does membrane penetration occur? Or it is just merely pore formation? And how can this depolarize the mitochondrial membrane and disrupts bioenergetic?
Answer: The MD simulation runs were done for 20 ns (lines 920-921), which is sufficient time for the cytotoxin to make initial contact with the membrane during the molecular dynamics simulations. In addition, this simulation showed that the cytotoxin specifically targets cardiolipin with higher affinity compared to other anionic phospholipids, which addressed our primary question in this MD study. The subsequent events in the cytotoxin’s interaction and its ability to penetrate the outer mitochondrial membrane and form pores depends on the concentration of cardiolipin, the number and type of basic amino acids residues located in the N-terminal domain of the cytotoxin, its interaction with other neighboring anionic and neutral phospholipids as well as other physico-chemical conditions (charge of the membrane, concentration of protons and pH). These conditions were discussed in lines 929-935 of the revised version of the manuscript. Regarding the mechanism of cytotoxin-triggered mitochondrial membrane depolarization and disruption of bioenergetics, this process is a multistep event and remains the subject of intense investigation. Based on several research studies that we highlighted in our review, we suggest that cytotoxins modulate mitochondrial bioenergetics mainly by inducing compartmentalization in crista membranes and by forming inverted micelles that serve as nanocarriers of protons to help shuttle protons from the F0 sector of the ATP synthase to respiratory complexes.
Reviewer 2 Report
Comments and Suggestions for Authors
In this manuscript, authors provided a comprehensive review of mitochondrial energetics, various aspects of ETC and ATP, and how cobra venom cardiotoxins can be used as a tool to solve some of the controversies regarding the mechanisms. The review is quite extensive but the content is essential and relevant to the topic in review. There are a few minor comments for authors' consideration, as follows:
1. The majority part of the paper (more than 70% perhaps?) dealt mainly with the basis or background of mitochondrial energetics, ETC / ATP, cardiolipin binding etc.. while cardiotoxins input seems to be secondary and suggestive. It is not denying the relevance and pharmacological implication of cardiotoxins in this regard, but perhaps the title and abstract can be modified to reflect the content.
2. Line 47: "... are the most widely studied cardiotoxins from snake venom. " -- this is controversial and questionable, as there are many other cardiotoxins/cytotoxins from other species that have been extensively studied. Suggest to rephrase or provide more evidence to support this statement.
3. Line 66-68: It should be noted that the activities are differential and dependent of the species origin. Different cardiotoxins have varying sequence structures which affect the mechanism and potency. Perhaps can refer to some comparative studies which looked at different CTX from different species, example Naja atra, Naja kaouthia and so on.
4. Figure 1 caption: provide the full names for abbreviations CL, PC, and explain briefly the color indication of these head groups.
Author Response
- The majority part of the paper (more than 70% perhaps?) dealt mainly with the basis or background of mitochondrial energetics, ETC / ATP, cardiolipin binding etc.. while cardiotoxins input seems to be secondary and suggestive. It is not denying the relevance and pharmacological implication of cardiotoxins in this regard, but perhaps the title and abstract can be modified to reflect the content.
Answer: In response to the reviewer’s comment, we have modified the title and made a few changes in the abstract accordingly to fit the content of the review paper. The new title of the manuscript is “Cobra Venom Cytotoxins as a Tool for Proving Mechanisms of Mitochondrial Energetics and Understanding Mitochondrial Membrane Structure”. In our review manuscript, we deemed it important to give the readers extensive background on the chemiosmotic theory in order to highlight some controversies and unsolved discrepancies in the model and how cardiotoxins, which we term cytotoxins in the revised version of manuscript, in the subsequent sections of the review contributed to reconciling some of those controversies and increasing our understanding of mitochondrial bioenergetics. For instance, we emphasize the breakthrough observations from other research studies that showed that the increase in mitochondrial ATP synthesis induced by non-bilayer structures generated by cardiotoxin treatment of isolated mitochondria revealed an important molecular mechanism by which mitochondria regulate bioenergetics (e.g. how non-bilayer structures induced by cytotoxins regulate ATP synthesis by binding to cardiolipin in isolated membrane model systems, presumably using a proton wire mechanism). Therefore, to appreciate the critical input of cardiotoxins, the latest views on debated issues in bioenergetics are very important, not only for progress in this field but also for the development of cardiotoxin applications in the treatment of age-related pathologies by means of mitochondrial rejuvenation.
- Line 47: "... are the most widely studied cardiotoxins from snake venom. " -- this is controversial and questionable, as there are many other cardiotoxins/cytotoxins from other species that have been extensively studied. Suggest to rephrase or provide more evidence to support this statement.
Answer: As per the reviewer’s comment, we have changed the phrase “the most widely studied” to “well characterized” on line 53 of the revised manuscript.
- Line 66-68: It should be noted that the activities are differential and dependent of the species origin. Different cardiotoxins have varying sequence structures which affect the mechanism and potency. Perhaps can refer to some comparative studies which looked at different CTX from different species, example Naja atra, Naja kaouthia and so on.
Answer: Thank you for this astute observation and we agree with the reviewer’s comment which we addressed in lines 68-77 of the revised review manuscript.
- Figure 1 caption: provide the full names for abbreviations CL, PC, and explain briefly the color indication of these head groups.
Answer: Thank you for this observation. We have made this minor edits in the figure caption of figure 1 as highlighted in yellow and by TrackChanges in the revised Word document.
Reviewer 3 Report
Comments and Suggestions for Authors
This review describes the research on the effects of snake venom cardiotoxins on mitochondrial membrane and energy metabolism, and summarizes the possible mechanisms involved. At the same time, the possible relationship between snake venom cardiotoxins and potential neurodegenerative disease related molecules was also explored. The paper has certain guiding significance for understanding the structure and function of snake venom cardiotoxins. However, there are still some issues with the paper:
1. For professional toxin research journals, the second part of the review of cell bioenergetics may be too long. I think this section only needs to briefly introduce the mainstream theories and working models of cell bioenergetics, and point out the existing problems to lay the theoretical foundation for the next section.
2. In the fourth part of the paper, Neurogenerative Pathology and Cardioxins, the author only described the similarities between the relevant molecules in neurogenerative pathology and cardiotoxins. From the perspective of snake venom cardiotoxins, there is no very definite experimental evidence as an amyloidogenic sequence trap. Suggest the author to weaken the description in this section and briefly describe it in future research directions.
3. In the fourth part of the paper: Neurogenerative Pathology and Cardioxins, the description of “Interaction of Cardiotoxin with Model Outer and Inner Mitochondrial Membranes” mainly describes the effects of snake venom cardiotoxins on mitochondrial membrane and mitochondrial energy metabolism. Suggest placing this section in the third section. At present, many studies have shown that the main effects of snake venom components on the nervous system are snake venom neurotoxins, phospholipase A2 and ion channel toxins, etc. Considering the extensive effects of snake venom cardiotoxins on cell membranes, it can be inferred that snake venom cytotoxic toxins may act on various nerve cells. Although reference 36 and other studies have used neural system related cells such as SH-SY5Y as models for research, it is still unclear how snake venom affects neurogenic pathologies. At least, whether snake venom can pass through the blood-brain barrier is a huge question. Of course, it cannot be ruled out that snake venom cell toxins can affect peripheral nerve fibers, myelin sheaths, and synapses, causing neurotoxicity. However, as a review of professional research field, some descriptions without sufficient experimental evidence may bring confusion and even mislead to the authors.
4. Based on the above issues, it is also recommended that the author modify the title of the paper appropriately and weaken “neurogeneration”.
Comments on the Quality of English LanguageNone
Author Response
- For professional toxin research journals, the second part of the review of cell bioenergetics may be too long. I think this section only needs to briefly introduce the mainstream theories and working models of cell bioenergetics, and point out the existing problems to lay the theoretical foundation for the next section.
Answer: Thank you for this observation. We understand the reviewer’s point of view and have carefully reviewed the second section of the paper, deciding to cut several lines of text (186-196 in original version of manuscript) pertaining to very well-known knowledge about the structure and mechanics of how mitochondrial complexes facilitate electron flow from the original manuscript. However, while we recognize that the content in bioenergetics and oxidative phosphorylation can be perceived as long, we deem it that it is necessary to give the readers extensive background on the chemiosmotic theory in order to highlight some controversies and unsolved discrepancies in the model which sets the stage in the second half of the review manuscript to explain how cardiotoxins, which we term cytotoxins in the revised version of manuscript, contributed to reconciling some of controversial aspects of the model and thereby increase our understanding of mitochondrial bioenergetics (e.g. how non-bilayer structures induced by cytotoxins regulate ATP synthesis by binding to cardiolipin in isolated membrane model systems, presumably using a proton wire mechanism). For instance, we emphasize the breakthrough observations from other research studies that showed that the increase in mitochondrial ATP synthesis induced by non-bilayer structures generated by cardiotoxin treatment of isolated mitochondria revealed an important molecular mechanism by which mitochondria regulate bioenergetics. Hence, in this context, readers will appreciate the contribution that cardiotoxin research have contributed in reconciling these controversial issues of the model. Additionally, the remaining content aids in understanding how cardiotoxins are currently being used and their potential applications in further research. For instance, we mentioned how cardiotoxins can be used as tools to unravel molecular RedOx mechanisms, study the role that non-bilayer structures play in mitochondria for regulating mitochondrial membrane remodeling and function, and to develop novel pharmaceuticals aimed at treating cellular energy deficiencies by restoring mitochondrial energy production by modulating the ATP synthase at low concentrations.
- In the fourth part of the paper, Neurogenerative Pathology and Cardioxins, the author only described the similarities between the relevant molecules in neurogenerative pathology and cardiotoxins. From the perspective of snake venom cardiotoxins, there is no very definite experimental evidence as an amyloidogenic sequence trap. Suggest the author to weaken the description in this section and briefly describe it in future research directions.
Answer: Thank you for this observation, we agree with the reviewer that, apart from our unpublished molecular dynamics data and the structural and biophysical similarities between cardiotoxins and amyloidogenic sequence traps, there is no direct experimental evidence that shows with high certainty that cardiotoxin sequences may serve as amyloidogenic sequence traps. While we agree that this is speculative, the similarity in the secondary structures between amnyloidogenic traps (Figure 9) and cytotoxins and biophysical characteristics gives some credence to the concepts that cytotoxins can act as amyloidogenic traps. Therefore, we agree with the reviewer to tone down and weaken the description of this section (two paragraphs consisting of lines 756-777) by explicitly stating that this is a though provocative research hypothesis and direction worth pursuing in the future as noted in lines 770-777 and 1073-1081 of the revised manuscript.
- In the fourth part of the paper: Neurogenerative Pathology and Cardioxins, the description of “Interaction of Cardiotoxin with Model Outer and Inner Mitochondrial Membranes” mainly describes the effects of snake venom cardiotoxins on mitochondrial membrane and mitochondrial energy metabolism. Suggest placing this section in the third section. At present, many studies have shown that the main effects of snake venom components on the nervous system are snake venom neurotoxins, phospholipase A2and ion channel toxins, etc. Considering the extensive effects of snake venom cardiotoxins on cell membranes, it can be inferred that snake venom cytotoxic toxins may act on various nerve cells. Although reference 36 and other studies have used neural system related cells such as SH-SY5Y as models for research, it is still unclear how snake venom affects neurogenic pathologies. At least, whether snake venom can pass through the blood-brain barrier is a huge question. Of course, it cannot be ruled out that snake venom cell toxins can affect peripheral nerve fibers, myelin sheaths, and synapses, causing neurotoxicity. However, as a review of professional research field, some descriptions without sufficient experimental evidence may bring confusion and even mislead to the authors.
Answer: While we understand that rearranging the two sections may increase the fluidity of the concepts, we prefer to converge the two themes as one section and renaming it as “Mechanisms of mitochondrial targeting: implications to cytotoxin-induced apoptosis in neuronal systems” which will maintain the organization of these two topics. The rationale behind this edit is that both themes fall within mitochondrial biology. The third section focuses on how cardiotoxins affect mitochondrial bioenergetics, whereas the subsection in the fourth section focuses on mitochondrial targeting by cardiotoxins in neuronal systems, leading to apoptosis and necrosis. Finally we agree that cardiotoxins may differentially affect other neuronal populations based on phospholipid composition of neuronal membranes including unmyelinated vs. myelinated nerves and may differentially affect glia vs. oligodendrocytes which warrants future studies. In addition, we agree that there is currently no knowledge on the ability of cytotoxins to penetrate the blood brain barrier as noted by the reviewed. Hence, based on these recommendations, we added additional language to the discussion in lines the discusses these possibilities on lines 1005-1015 of the revised manuscript.
- Based on the above issues, it is also recommended that the author modify the title of the paper appropriately and weaken “neurodegeneration”.
Answer: Thank you for this observation. We agree with the reviewer, and we have modified the title of the paper. We have addressed the other above issues in our answers to points 2 and 3.
Round 2
Reviewer 3 Report
Comments and Suggestions for Authors
The authors have made great efforts to mend the manuscript and answered most of the questions. I think it is acceptable for Toxins now.